# RAPA: Recursively Aligned Pathway Adaptation of Large Language Models

## Abstract

Parameter-Efficient Fine-Tuning (PEFT) adapts large language models (LLMs) by training only a small fraction of parameters. Adapter-based approaches reduce compute per step but introduce practical overhead from the additional adapter path (e.g., extra kernel launches and activation storage). Adapter-free approaches avoid this structural overhead by directly updating pretrained weights; however, per-layer random index selection can fragment the trainable subspace, attenuating gradient flow and limiting accuracy. We propose **Recursively Aligned Pathway Adaptation (RAPA)**, an adapter-free PEFT method that forms index-consistent pathways through depth. RAPA follows two principles: (i) selecting balanced submatrices that maximize the number of weights alignable across layers, and (ii) recursively aligning these indices across layers and residual connections. In experiments, RAPA matches or surpasses strong PEFT baselines across most benchmarks while preserving adapter-free efficiency with minimal memory and compute overhead. Code is available at `https://anonymous.4open.science/r/rapa`.

## 1 Introduction

Large Language Models (LLMs) now match or surpass expert-level performance in many natural language processing tasks (Zhao et al., 2024b; Team et al., 2024; Dubey et al., 2024; DeepSeek-AI et al., 2025; Jiang et al., 2024a; Chowdhery et al., 2023). In particular, to specialize these LLMs for downstream applications, fine-tuning is commonly employed to adjust their parameters using task-specific data (Wolpert & Macready, 1997). However, fine-tuning these large-scale models for specific tasks demands substantial computational resources and memory (Brown et al., 2020; Achiam et al., 2023; Kaplan et al., 2020), posing significant challenges to practical applications.

To address these challenges, a wide range of Parameter-Efficient Fine-Tuning (PEFT) schemes have been proposed, aiming at reducing computational costs and memory footprints required during fine-tuning. Representative techniques such as Adapter Tuning (Houlsby et al., 2019; Pfeiffer et al., 2020), Low-Rank Adaptation (LoRA) (Hu et al., 2022) and its variants (Liu et al., 2024; Wu et al., 2024a) significantly improve efficiency by freezing pre-trained weights and introducing a small number of trainable adapter parameters.

In theory, adapters introduce only a small amount of additional computation, but they can result in significant training-time overhead in practice; the base and adapter paths are executed sequentially, and computing gradients for adapter parameters typically requires retaining full input activations, which can increase kernel launches and activation memory. By contrast, adapter-free approaches do not require the adapter modules and hence may avoid these structural sources of overhead, at the expense of different trade-offs: BitFit (Zaken et al., 2021b) and LayerNorm tuning (Zhao et al., 2024a) prioritize simplicity and stability over capacity, and PaCA (Woo et al., 2025) selects trainable coordinates at the layer level without explicit cross-layer structure, limiting expressiveness and slowing down convergence in some settings. Our study focuses on characterizing these trade-offs and designing an improved PEFT approach.

In this paper, we first analyze the sources of training overhead in widely used adapter-based methods. We then introduce **Recursively Aligned Pathway Adaptation (RAPA)**, an adapter-free PEFT algorithm that selects and aligns indices across all layers so that the updated weights form a consistent gradient pathway (Figure 1(a)). By aligning the trainable coordinates through depth (and, when applicable, choosing shape-balanced submatrices), RAPA aims to enhance representational capacity

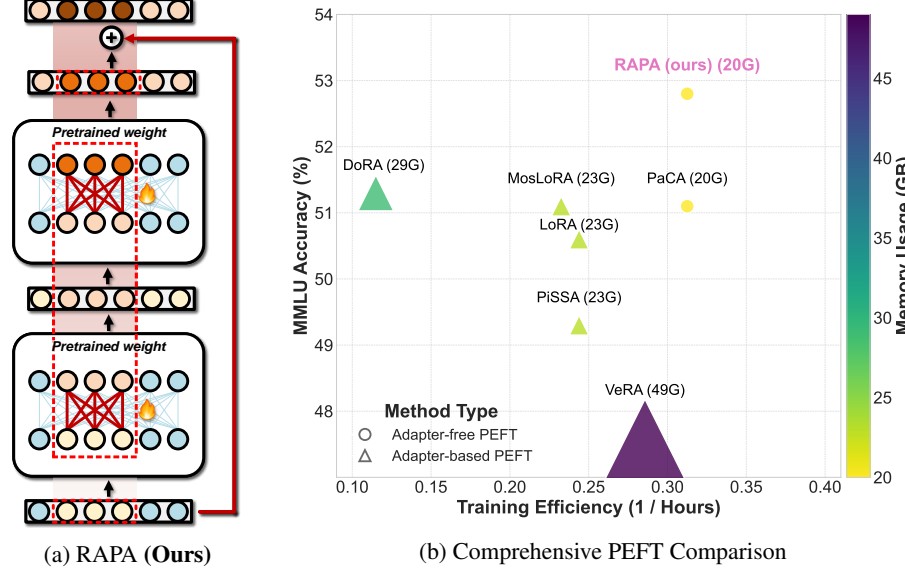

(a) RAPA **(Ours)**      (b) Comprehensive PEFT Comparison

Figure 1: Performance comparison of proposed RAPA method with other PEFTs. **(a)** 5-shot MMLU accuracy compared with other PEFT methods on different tuning parameter budgets. **(b)** Comprehensive comparison of PEFT methods across accuracy, efficiency, and memory usage, highlighting RAPA's overall superiority.

and stabilize gradient flow while avoiding both adapter-induced overhead and per-layer re-sampling costs.

We evaluate RAPA across diverse models and tasks. Specifically, we conduct 5-shot MMLU fine-tuning (Hendrycks et al., 2020) (Figure 1(b)), commonsense reasoning benchmarks, and instruction tuning followed by MT-Bench evaluation. All experiments are run on a single NVIDIA A100 80 GB GPU under matched trainable-parameter budgets and comparable optimization settings to ensure fair comparison with prior PEFT baselines. Full datasets, hyperparameters, and evaluation protocols are provided in the experimental setup for reproducibility.

## 2 BACKGROUND AND MOTIVATION

### 2.1 CHALLENGES IN ADAPTER-BASED PEFT

Fine-tuning LLMs typically incurs significant memory and computation costs. This is mainly because back-propagation requires storing all intermediate activations and updating a massive set of parameters. These requirements hinder practical deployment of LLMs, especially on resource-constrained hardware.

To address these challenges, PEFT methods aim to reduce the number of trainable parameters while maintaining task accuracy. One widely used approach is LoRA (Hu et al., 2022), which introduces additional rank-constrained matrices to the existing pre-trained weights as below:

$$W = W_0 + \Delta W = W_0 + s \cdot BA$$

where $A \in \mathbb{R}^{r \times d}$, $B \in \mathbb{R}^{d \times r}$, and $s$ is a scaling factor. While LoRA reduces the number of trainable parameters, it still requires storing all input activations in memory, as the gradients for the low-rank adapter matrices must be computed with respect to the original input activation. In addition, LoRA introduces sequential computation steps and hence suffers from latency overhead during training; the forward propagation of LoRA proceeds sequentially through the pre-trained weights and the low-rank adapter, which hinders hardware-level parallelism. It was reported that LoRA incurs approximately 33% training-time overhead compared to full fine-tuning, as the forward pass must compute both the pre-trained weight and the low-rank adapter in series rather than in parallel (Hu et al., 2022; Woo et al., 2025). In addition, its low-rank structure limits representational capacity on complex tasks.

## 2.2 Approaches to Weight Selection in Adapter-Free PEFT

The central challenge in adapter-free PEFT is selecting which subset of weights to update. Existing strategies range from simple random selection, which prioritizes speed and simplicity (Woo et al., 2025), to importance-based approaches that score weights via gradient magnitudes or activation sensitivities. However, these importance-driven methods typically require a costly precalibration step to compute these scores, introducing additional overhead and potential sensitivity to the calibration data (He et al., 2025).

In this work, we propose a *calibration-free, structure-aware selection* strategy that aligns trainable coordinates across layers (Section 3.2). By leveraging the inherent model architecture, our method avoids precomputation overhead while yielding tight seed-to-seed variability (Appendix A) and achieving high performance across tasks.

## 3 Our work

In this section, We first characterize the training-time overhead of adapter-based PEFT. We then introduce *Recursively Aligned Pathway Adaptation (RAPA)*, a calibration-free, structure-aware weight selection method. The alignment is applied *recursively* across all layers—mirroring the top-down flow of backpropagation—to form coherent pathways. This is achieved by tuning balanced square submatrices of pretrained weights, enforcing cross-layer consistency in the chosen coordinates, and maximizing shared indices across layers under a fixed parameter budget. In experiments, RAPA achieves faster convergence and higher accuracy while preserving adapter-free training efficiency and avoiding calibration or per-layer resampling overhead.

### 3.1 Revisiting the Overhead of Adapter-based PEFT

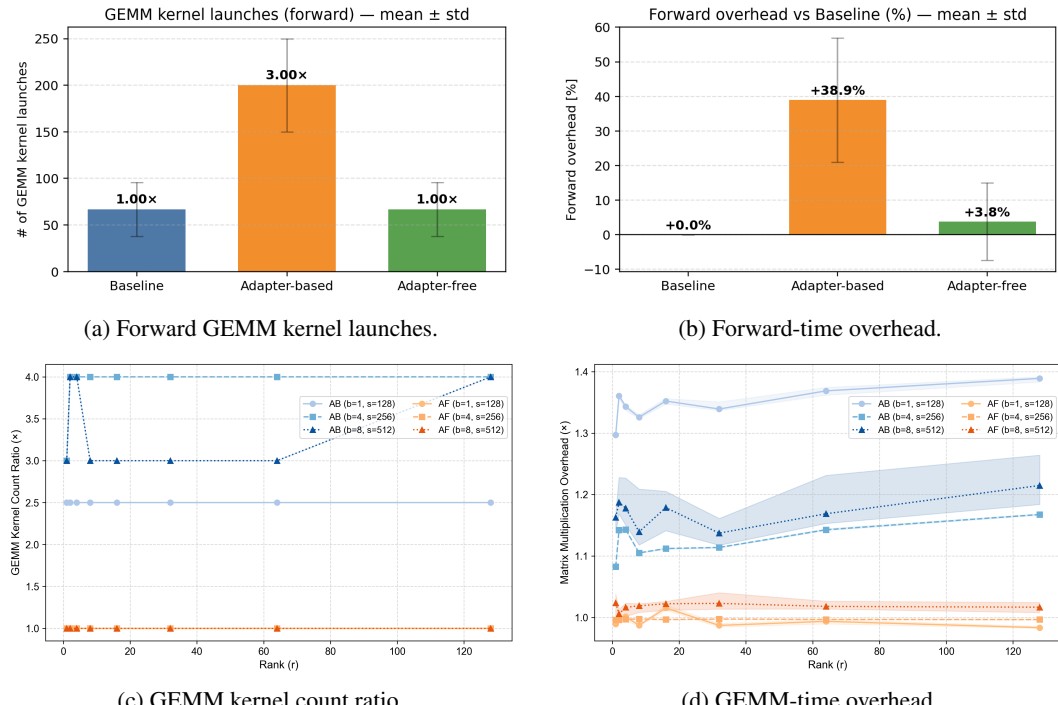

(a) Forward GEMM kernel launches.

(b) Forward-time overhead.

(c) GEMM kernel count ratio.

(d) GEMM-time overhead.

Figure 2: **(a)** Forward GEMM kernel launches. Adapter-based (AB) issues $\sim 3\times$ more kernels than the baseline. **(b)** Forward-time overhead. AB increases latency by $\sim 40\%$, whereas adapter-free (AF) closely matches baseline. **(c)** Kernel-count ratio (AB/baseline). The overhead pattern is largely insensitive to adapter rank $r$. **(d)** GEMM-time ratio (AB/baseline). Overhead depends more on batch size $b$ and sequence length $s$.

To motivate our adapter-free (AF) approach, we conduct a controlled microbenchmark to isolate the sources of overhead in adapter-based (AB) methods. On a single linear layer ($4096{\times}4096$, `bfloat16`) using an NVIDIA A100 (80 GB) GPU, we compare an AB setup (LoRA (Hu et al., 2022)) against an AF setup and a frozen baseline, keeping the trainable parameter budget identical for both AB and AF. We measure throughput with CUDA events and collect CUDA-kernel statistics using `torch.profiler`; results are aggregated over multiple trials (median, with 95% bootstrap confidence intervals).

As illustrated in Figure 2(a), the AB method substantially increases the number of General Matrix Multiplication (GEMM) kernel launches—by over $3\times$ on average, which corresponds to a forward-time overhead of $\sim 40\%$ (Figure 2(b)). In contrast, AF keeps the kernel count essentially unchanged and incurs only a small overhead ($\sim 4\%$).

These results indicate that the dominant inefficiency is not added FLOPs but the proliferation of small, sequential kernel launches that underutilize the GPU (Shi et al., 2016). In Figure 2(c), the kernel-count ratio is largely insensitive to adapter rank $r$. By contrast, Figure 2(d) shows that the GEMM-time ratio depends more on batch size $b$ and sequence length $s$. Simply scaling $b$ or $s$ to improve utilization is often impractical due to activation-memory growth in deep models. This structural overhead of separate adapter modules motivates our method RAPA, which aims to retain high performance without incurring this computational burden. We further confirm that this trend holds consistently on both a smaller model and a larger model; the corresponding overhead measurements are reported in Appendix I.

## 3.2 RAPA: RECURSIVELY ALIGNED PATHWAY ADAPTATION

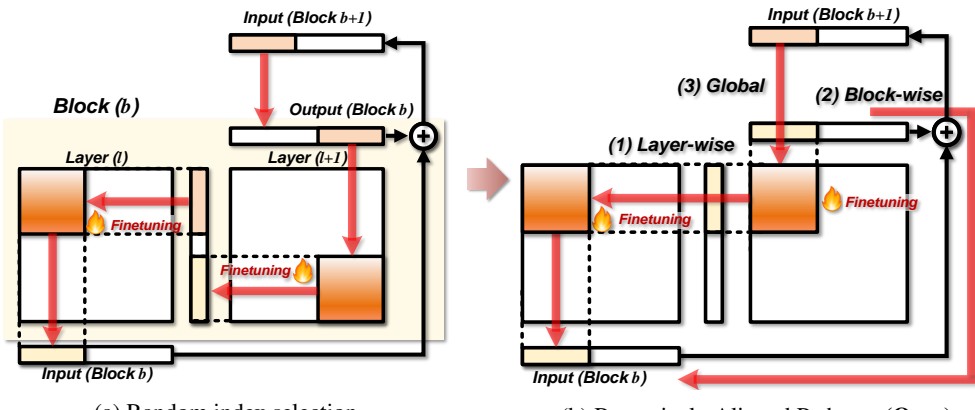

(a) Random index selection      (b) Recursively Aligned Pathway (**Ours**)

Figure 3: Comparison between the naive random index selection and the proposed recursively aligned fine-tuning. **(a)** The naive method randomly selects indices for fine-tuning, resulting in partially frozen weights that disrupt gradient flows and lead to inefficient convergence. **(b)** The proposed approach selects square-shaped submatrices for fine-tuning, structurally aligning connections across layers to maintain continuous gradient propagation and accelerate loss convergence.

### 3.2.1 BALANCED SUBMATRIX SELECTION

Let $W \in \mathbb{R}^{d \times d}$ be a pretrained weight matrix. LoRA has rank $r$, updating $2 \times r \times d$ parameters. We keep the same parameter budget but reshape it into a $k \times k$ square block:

$$\text{\# Trainable Params: } k^2 = 2\,r\,d \implies k = \sqrt{2\,r\,d}, \qquad \text{rank}_{\text{balance}} = k > r \quad (\text{when } r < d). \quad (1)$$

We propose selecting a square-shaped block of weights of size $k \times k$ when tuning $k^2$ weights (Eq. 1). Thus the rank increases from $r$ to $\sqrt{2\,r\,d}$ without affecting the rest of the matrix. This rank expansion typically endows the fine-tuned subspace with greater expressive capacity, enabling the adapter to capture more complex task-specific patterns (Jiang et al., 2024b).

Since we still update exactly $2\,r\,d$ parameters, the per-step FLOPs and optimizer memory are unchanged; only their 2-D locations differ. Unlike column-only or row-only selections, this square-

shaped selection balances input and output dimensions and leads to an increased rank of the fine-tuned subspace. As a result, it provides richer expressiveness.

Experimental results in Section 4.1 demonstrate a steady improvement in accuracy as the row-to-column ratio approaches a balanced state. Optimal scores are achieved within a range closely centered around, but not precisely at, 1:1 ratio. This trend confirms that increasing the rank of the fine-tuned subspace is a critical factor in improving fine-tuning accuracy. Furthermore, balanced-shaped selection maximizes the opportunity for index alignment with adjacent layers, thereby reinforcing coherent cross-layer learning pathways.

### 3.2.2 STRUCTURAL INDEX ALIGNMENT: OPTIMIZING GRADIENT FLOW

RAPA enhances training dynamics by structurally aligning the indices—the selected row and column positions—of fine-tuned weights across layers. This strategy promotes coherent gradient propagation by reducing discontinuities, which accelerates convergence.

Throughout this paper, we use the term *block* to denote a Transformer sub-module, namely an attention block or a feed-forward network (FFN) block. For notational clarity, we model each block as **two consecutive linear layers** whose input and output are linked by a residual connection (He et al., 2016b;a).

In this section, we describe three types of index alignment strategies:

- **Layer-wise Alignment:** Align output indices of a layer with input indices of the next.
- **Block-wise Alignment:** Align indices across both linear layers and the residual connection inside each attention or FFN block.
- **Global Alignment:** Align indices consistently across multiple residual-connected blocks.

**Method.** An overview of the index alignment method is provided in Figure 3. Since we update only a $k \times k$ sub–matrix per layer, *which weights are selected* determines whether the back-propagated gradient can continue propagating through trainable paths or is blocked by frozen weights. The core idea is therefore simple:

(i) Choose a fixed index set for the trainable subspace.

(ii) Apply it consistently across all residual-compatible layers.

This single decision simultaneously enforces *layer*, *block*, and *global* alignment. To implement this strategy, we begin by selecting a fixed index set $I, J \subset \{1, \ldots, d\}$ of size $k$ from the hidden dimension $d$. In each transformer layer $l$, we update only the square of the weight matrix $W_{I,J}^{(l)}$.

$$x_I^{(l+1)} = W_{I,J}^{(l)} x_J^{(l)}, \quad x_J^{(l+2)} = x_J^{(l)} + W_{J,I}^{(l+1)} x_I^{(l+1)} \tag{2}$$

As shown in Eq. 2, the layer-wise alignment is achieved by selecting the same index set across two consecutive layers, highlighted in blue. The red-colored indices represent block-wise alignment, which matches the index sets of the input at layer $l$ and the output at layer $l+2$, taking into account the residual connection within the block.

We express global alignment across residual-connected blocks as

$$x_J^{(b+1)} = x_J^{(b)} + W_{J,I}^{(b+1)} \left( W_{I,J}^{(b)}, x_J^{(b)} \right) = x_J^{(b)} + \mathcal{F} \left( W_{J,J}^{(b)}, x_J^{(b)} \right) \tag{3}$$

$$\implies x_J^{(b+m)} = x_J^{(b)} + \sum_{j=0}^{m-1} \mathcal{F} \left( W_{J,J}^{(b+j)}, x_J^{(b+j)} \right) \tag{4}$$

Eq. 3 describes the forward propagation of a block indexed by $b$, and its generalization to $m$ consecutive blocks in Eq. 4, where $\mathcal{F}(\cdot)$ denotes the transformation through the two linear layers within each block. These alignment strategies enhance direct gradient propagation through residual shortcuts and inter-layer connections.

As a result, Figure 4 shows that the training loss converges noticeably faster when the trainable weight indices are *aligned*. As demonstrated in Table 1, alignment not only accelerates convergence but also

improves final accuracy. To confirm the robustness of this trend, we present the detailed training loss curves across multiple random seeds in Appendix E. The full strategy is summarized in Appendix G.

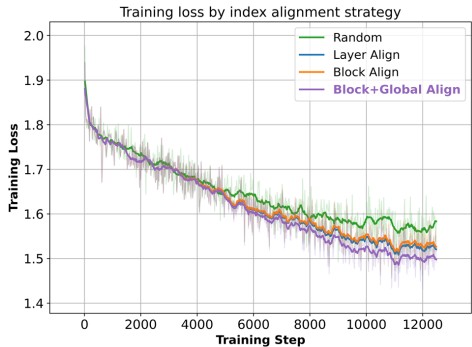

Figure 4: Comparison of training loss curves for index alignment strategies.

Table 1: Effect of index alignment strategies on 5-shot MMLU accuracy.

| Method | Hums. | STEM | Social. | Other | Avg. |
|---|---|---|---|---|---|
| Random | 48.2 | 41.3 | 60.2 | 59.4 | 51.7 |
| Layer Align | 49.3 | 42.1 | 59.8 | 58.4 | 52.0 |
| Block Align | 49.4 | 41.5 | 59.9 | 57.9 | 51.8 |
| **Block+Global** | 49.3 | 42.1 | 60.3 | 59.1 | **52.3** |

**Why alignment works.** We provide a simple back-propagation analysis to illustrate how the index alignment method facilitates more effective loss convergence. The chain rule for any trainable element of $W_{I,J}^{(l)}$ yields

$$\frac{\partial \mathcal{L}}{\partial W_{I,J}^{(l)}} = \frac{\partial \mathcal{L}}{\partial x_J^{(l+2)}} \, W_{J,I}^{(l+1)} \, x_J^{(l)}. \tag{5}$$

As shown in Eq. 5, when index alignment includes $W_{J,I}^{(l+1)}$ in the trainable subspace, its update at layer $l+1$ is immediately reflected in the gradient of layer $l$. This leads to more consistent gradient directions and faster convergence.

We further strengthen this connection by choosing the same set $J$ for both the block input and the output connected via the residual path. This yields

$$\frac{\partial \mathcal{L}}{\partial x_J^{(l)}} = \frac{\partial \mathcal{L}}{\partial x_J^{(l+2)}} \left( 1 + W_{J,I}^{(l+1)} W_{I,J}^{(l)} \right) \tag{6}$$

where the identity term in Eq. 6 arises from the residual shortcut. The loss gradient at the block output $x_J^{(l+2)}$ is directly reflected in the gradient of the block input $x_J^{(l)}$ through the residual connection. As a result, the model can respond to the loss signal more quickly, leading to faster convergence.

Finally, we propagate the alignment globally across blocks by fixing the index set $J$ and reusing it as both the input and residual-output weights in every residual block.

$$\frac{\partial \mathcal{L}}{\partial x_J^{(b)}} = \underbrace{\frac{\partial \mathcal{L}}{\partial x_J^{(b+m)}}}_{\text{direct shortcut}} + \frac{\partial \mathcal{L}}{\partial x_J^{(b+m)}} \frac{\partial}{\partial x_J^{(b)}} \sum_{j=0}^{m-1} \mathcal{F}\left( W_{J,J}^{(b+j)}, x_J^{(b+j)} \right) \tag{7}$$

Back-propagating through $m$ such blocks results in Eq. 7. The equation above suggests that the gradient at block $b + m$ reaches block $b$ both directly through the residual shortcut and through the aligned subspace shared by all trained blocks (a detailed analysis is provided in Appendix B). Appendix C extends this pathway-alignment view to the attention layers, including Q/K alignment and the low-rank $QK^T$ interaction. Appendix D further shows that the same index-alignment mechanism applies to MoE residual blocks.

## 4 EXPERIMENTS

We conducted comprehensive experiments to evaluate the effectiveness and generalization capability of RAPA, along with the impact of critical design decisions. All experiments were performed on a

single NVIDIA A100 80GB GPU, using 16-bit mixed precision (Micikevicius et al., 2017) and the AdamW (Loshchilov & Hutter, 2017) optimizer unless otherwise specified.

We measured accuracy on the MMLU benchmark and on eight commonsense reasoning tasks (Clark et al., 2018; 2019; Zellers et al., 2019; Mihaylov et al., 2018; Bisk et al., 2020; Sap et al., 2019; Sakaguchi et al., 2020). Additionally, we conducted instruction tuning and evaluated the conversational capabilities of the model using the MT-Bench benchmark (Zheng et al., 2023). We also analyzed the effects of weight shape selection and index alignment strategies introduced in Section 3.2. Detailed experimental settings and hyperparameter configurations are provided in Appendix H.

## 4.1 Effects of Balanced Weight Selection and Alignment

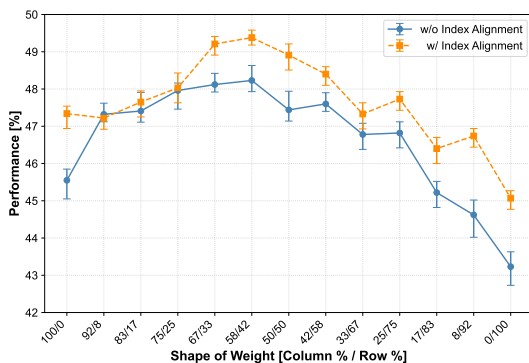

Figure 5: Effects of weight shape (row/column ratio) and index alignment on MMLU accuracy.

We investigated how the row-to-column ratio of trainable weights influences task accuracy and how this relationship changes after index alignment is applied. Figure 5 plots MMLU accuracy as a function of the selected row-to-column ratio on x-axis, comparing cases without index alignment (blue curve) and with index alignment (yellow curve). We fine-tuned the LLaMA2-7B model (Touvron et al., 2023) on the MMLU dataset for this analysis; the detailed experimental setup is provided in Appendix H (Table 9).

As shown in Figure 5, selecting a symmetric subset of weights yields higher task accuracy compared to extreme cases where only row or column weights are selected. Focusing on the random selection case (blue curve), a balanced (50/50) ratio achieves an accuracy of 47.4%, outperforming the extreme column-only case (100/0) and row-only (0/100) configurations by 1.8% and 4.2%, respectively.

This trend remains consistent when the **index alignment method** is applied: balanced selection again achieves the highest accuracy. Moreover, the aligned variant consistently outperforms the random baseline across all settings. In the balanced case, index alignment improves performance to 48.9%, 1.5% higher than the random selection method at 47.4%. Even in the column-only setting, alignment increases accuracy from 45.6% to 47.3%, and in the row-only setting, from 43.2% to 45.1%, confirming that alignment enhances performance even under asymmetric or suboptimal weight configurations.

## 4.2 Fine-tuning for Downstream Task

We validate the task accuracy and efficiency of RAPA using the MMLU 5-shot benchmark, which evaluates complex reasoning across 57 academic subjects. RAPA was compared against several PEFT baselines, including LoRA (Hu et al., 2022), DoRA (Liu et al., 2024), MosLoRA (Wu et al., 2024a), as well as PaCA (Woo et al., 2025), using the MMLU dataset (Hendrycks et al., 2020). All PEFT schemes were evaluated on both LLaMA2-7B and LLaMA2-13B. For fair comparisons, the LoRA baseline was configured with a rank of 128, and RAPA was adjusted to match the trainable parameter count. The results from additional experiments across various rank settings are reported in Appendix F, and the detailed configurations are provided in Appendix H.

On the MMLU benchmark, RAPA consistently outperforms other PEFT methods (Table 2). It achieves higher accuracy than PaCA (+0.7 % on LLaMA2-7B, +0.6 % on LLaMA2-13B) with

Table 2: Comparisons of memory usage (Mem), training time (Time), and 5-shot accuracy on MMLU dataset when fine-tuning the LLaMA2-7B and LLaMA2-13B models using various PEFT algorithms.

| Model | Method | # Params | Mem | Time | Accuracy (%) | | | | |
|---|---|---|---|---|---|---|---|---|---|
| | | | | | Hums. | STEM | Social. | Other | Avg. |
| LLaMA2-7B | Baseline | - | - | - | 43.6 | 37.1 | 51.4 | 53.0 | 45.9 |
| | LoRA | 319.8M | 53.4G | 3.3h | 50.6 | 43.3 | 61.9 | 59.5 | 53.4 |
| | DoRA | 321.1M | 75.3G | 6.2h | 51.8 | 43.1 | 61.9 | 60.1 | 53.9 |
| | MosLoRA | 323.5M | 53.4G | 3.5h | 50.7 | 42.4 | 60.5 | 58.2 | 52.6 |
| | PaCA | 348.1M | 41.7G | 2.8h | 50.7 | 43.3 | 63.5 | 60.1 | 53.9 |
| | **RAPA (Ours)** | 319.6M | 41.5G | 2.9h | **51.4** | **44.4** | **64.3** | **60.2** | **54.6** |
| LLaMA2-13B | Baseline | - | - | - | 53.2 | 44.2 | 62.8 | 60.8 | 55.0 |
| | LoRA | 500.7M | 60.3G | 5.9h | 56.4 | 47.5 | **67.9** | 64.2 | 58.6 |
| | MosLoRA | 505.3M | 60.6G | 6.1h | 55.5 | 46.0 | 67.4 | 64.2 | 57.9 |
| | DoRA | 502.8M | 80.3G | 12.0h | 57.0 | 47.8 | 67.6 | 64.6 | 58.9 |
| | PaCA | 545.3M | 52.7G | 5.1h | 57.2 | 47.5 | 67.5 | 65.1 | 59.0 |
| | **RAPA (Ours)** | 501.5M | 52.9G | 5.2h | **57.9** | **48.1** | 67.6 | **65.9** | **59.6** |

similar efficiency, and surpasses LoRA and MosLoRA in both accuracy and training speed. While DoRA reaches competitive accuracy, it exhibits significantly longer training time, highlighting the computational efficiency of RAPA. Among all compared methods, RAPA achieves the highest average accuracy on both models, demonstrating the effectiveness of the structured weight selection and index alignment strategies proposed in this work.

## 4.3 COMMONSENSE REASONING

Table 3: Comparison of PEFT methods on commonsense reasoning benchmarks.

| Method | Mem | Time | ARC-c | ARC-e | BoolQ | HellaS. | OBQA | PIQA | SIQA | WinoG. | Avg. |
|---|---|---|---|---|---|---|---|---|---|---|---|
| Baseline | - | - | 43.3 | 74.6 | 77.7 | 76.0 | 44.2 | 79.1 | 46.1 | 69.1 | 63.8 |
| LoRA | 53.2G | 4.8h | 43.0 | 72.6 | 80.6 | 67.8 | 44.6 | 76.6 | 54.9 | 74.3 | 64.3 |
| DoRA | 75.3G | 8.8h | 52.1 | 80.0 | 73.7 | 76.4 | 49.0 | 80.4 | 58.4 | 82.0 | 69.0 |
| MosLoRA | 53.4G | 5.0h | 53.0 | 81.2 | 70.7 | 76.3 | 50.2 | 80.8 | 58.8 | 82.1 | 69.1 |
| PaCA | 41.5G | 4.0h | 51.0 | 78.5 | 73.7 | 75.4 | 48.8 | 80.2 | 57.5 | 80.3 | 68.2 |
| **RAPA (Ours)** | 42.1G | 4.1h | **53.1** | **81.1** | **76.4** | **77.1** | **49.8** | **80.4** | **58.9** | **80.8** | **69.7** |

We further evaluated RAPA on eight canonical commonsense reasoning tasks (ARC Challenge, ARC Easy, BoolQ, HellaSwag, OpenbookQA, PIQA, Social IQA, and Winogrande), comparing its accuracy against LoRA, DoRA, MosLoRA, and PaCA. All experiments were conducted on the LLaMA2-7B model and we fine-tuned it on a commonsense reasoning dataset consisting of 170k samples. Detailed hyperparameters are listed in Appendix H.

Table 3 reports that RAPA achieves the highest macro-average accuracy of 69.7%, outperforming the strongest adapter-based baseline, DoRA (69.0%), by 0.7% and the strongest adapter-free baseline, PaCA (68.2%), by 1.5%.

Despite higher accuracy, RAPA keeps the computational footprint essentially flat: 42.1 GB peak GPU memory usage and 4.1 hour of wall-clock time, nearly identical to PaCA (41.5GB and 4.0 hour) and approximately 20% lower than those of LoRA and MosLoRA (53 GB), while requiring less than half the GPU time compared to DoRA (8.8 hours). Since RAPA remains adapter-free, its per-step FLOPs and optimizer state size are unchanged relative to PaCA.

These results reinforce the findings from the experiments on MMLU and underscore two key benefits of combining structured square selection with cross-layer index alignment: (i) a higher rank under a fixed parameter budget, which expands the hypothesis space sufficiently to model diverse commonsense reasoning tasks, and (ii) uninterrupted gradient flow across layers, which accelerates convergence without incurring additional memory overhead. Together, these results demonstrate that principled subspace design, rather than simply increasing parameter count, is a key to generalization performance in low-rank, adapter-free fine-tuning.

## 4.4 INSTRUCTION TUNING

We evaluated RAPA on the Mistral-7B model, which was instruction-tuned for one epoch on the OASST1 dataset using a single NVIDIA A100 GPU. Performance was subsequently measured using the MT-Bench benchmark, which averages scores over two conversational turns. We used GPT-4o-mini as the evaluator for this benchmark (see Table 12 in Appendix H for further details).

As shown in Table 4, RAPA achieved the highest average score of 4.84. Notably, compared to PaCA, another adapter-free method with nearly identical training efficiency, outperforms PaCA by almost 0.9 points (4.84 vs. 3.96). This demonstrates that RAPA overcomes the performance limitations of prior adapter-free approaches while also outperforming strong adapter-based methods like LoRA (4.55) and DoRA (4.63). Furthermore, this high performance is achieved with remarkable efficiency, requiring less than half the training time of DoRA.

Table 4: Performance comparison of instruction tuning on Mistral-7B (Jiang et al., 2023).

| Model | Method | Time | Mem | # Params | Human | STEM | Role | Extract | Writing | Reason | Coding | Math | Avg. |
|---|---|---|---|---|---|---|---|---|---|---|---|---|---|
| | Baseline | - | - | - | 1.85 | 3.20 | 2.95 | 3.15 | 2.50 | 1.70 | 2.30 | 1.40 | 2.38 |
| | LoRA | 47m | 52G | 168M | 6.40 | 5.75 | 4.35 | **5.20** | 5.25 | 4.20 | 3.30 | 1.95 | 4.55 |
| Mistral-7B | DoRA | 93m | 69G | 169M | **6.70** | 5.60 | 4.30 | 4.40 | 5.75 | **4.50** | **3.85** | 1.95 | 4.63 |
| | MosLoRA | 48m | 52G | 169M | 5.80 | **6.05** | 4.85 | 5.00 | 5.00 | 3.75 | 3.65 | 2.00 | 4.51 |
| | PaCA | 41m | 47G | 176M | 5.35 | 5.45 | 4.15 | 4.50 | 4.55 | 2.55 | 3.20 | 1.95 | 3.96 |
| | **RAPA (Ours)** | 42m | 48G | 168M | **6.70** | 5.82 | **5.65** | **5.20** | **5.85** | 4.20 | 3.22 | **2.05** | **4.84** |

## 5 RELATED WORK

**Parameter-Efficient Fine-tuning (PEFT)** A range of PEFT methods have been proposed to adapt large pre-trained models to downstream tasks while minimizing the number of trainable parameters (Han et al., 2024; Zaken et al., 2021a; Liu et al., 2022). Early approaches introduced adapter modules within each transformer layer (Houlsby et al., 2019; Pfeiffer et al., 2020), inserting small bottleneck networks that are trained while keeping the backbone frozen. Although being effective, these methods incur both training and inference overheads due to the additional network components. To address the inference inefficiency, LoRA (Hu et al., 2022) was proposed, which employs low-rank decomposed matrices during training and merges them back into the original weights during inference. This scheme enables LoRA and its variants (Liu et al., 2024; Wu et al., 2024b) to achieve zero inference-time overhead while maintaining high accuracy. In parallel, side-network based approaches such as Side-Tuning and READ train a lightweight side branch in addition to a frozen backbone, combining the two predictions additively or via a gating mechanism (Zhang et al., 2020; Nguyen et al., 2023). Conceptually, these methods share the same additive nature as LoRA-style adapters in that they attach extra modules or forward paths around a frozen backbone.

**Parameter-efficient Column-wise Adaptation (PaCA)** PaCA (Woo et al., 2025) is a lightweight fine-tuning method that randomly selects and updates a subset of column-wise weights from the pre-trained model without introducing additional adapter modules. Unlike LoRA-based schemes that utilize additional trainable low-rank adapters, PaCA performs adapter-free fine-tuning, thereby reducing both training time and memory consumption. However, the use of random index selection without structural alignment across layers lead to suboptimal gradient propagation and degraded convergence behavior in some cases.

**Sparse Matrix Tuning (SMT)** SMT (He et al., 2025) suggests gradient-based importance estimation to selectively fine-tune substructures within the pre-trained model, demonstrating improved accuracy and faster convergence compared to conventional PEFT schemes. However, SMT relies on performing full fine-tuning over the entire model for a number of iterations to obtain importance scores, incurring substantial computational and memory overhead. Furthermore, since the importance-based tuning modifies a subset of the full model parameters directly, it necessitates storing the entire model after adaptation, rather than storing only the lightweight adapter modules. This characteristic limits the scalability of SMT in multi-task or multi-domain scenarios (Sheng et al., 2023; Li et al.,

2024; Zhao et al., 2024c; Zeng et al., 2025), where deploying only a specialized adapter for each task or domain is a key advantage of PEFT.

## 6 CONCLUSION

We presented RAPA, an adapter-free PEFT method that expands capacity and stabilizes optimization without added computational or memory cost. RAPA reallocates a fixed parameter budget to a balanced square submatrix and aligns the corresponding indices across residual-connected layers, yielding a higher rank of the weight update and coherent gradient pathways. The design is motivated by a simple backpropagation analysis and incurs no extra modules or calibration passes. Across MMLU, commonsense reasoning, and MT-Bench, RAPA improves accuracy and convergence under matched budgets while keeping time and memory near the adapter-free baseline. We summarize limitations and broader impacts in Appendices K and L.

## ETHICS STATEMENT

Our research aims to make the fine-tuning of large language models more computationally efficient and accessible. By reducing the resources required for parameter-efficient fine-tuning, our method, RAPA, can help democratize access to advanced AI technologies, enabling a wider range of researchers, startups, and organizations to develop specialized models for beneficial applications. We acknowledge that any technology that lowers the barrier to entry for powerful models also carries the risk of misuse. More efficient fine-tuning techniques could potentially be leveraged by malicious actors to generate harmful content, such as high-quality disinformation or spam, at a larger scale. Furthermore, while RAPA is a general optimization method, the models fine-tuned with our technique will inherit any social biases present in the base model and the fine-tuning data. We believe the benefits of enabling broader access for legitimate research and innovation are substantial. We encourage the community to continue developing robust safeguards and responsible deployment practices for fine-tuned models. We have conducted our work in full adherence to the ICLR Code of Ethics.

## REPRODUCIBILITY STATEMENT

We are committed to the full reproducibility of our research. All experiments were conducted with a rigorous methodology, ensuring fair comparisons across methods under identical trainable-parameter budgets. We report our findings transparently, detailing performance across all benchmarks. To enable other researchers to verify and build upon our work, we provide our source code in the supplementary material. Furthermore, all datasets used are publicly available, and the complete experimental configurations and hyperparameters for each experiment are detailed in Appendix H.

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

## A    ROBUSTNESS OF FINE-TUNING VS. INITIAL WEIGHT SENSITIVITY

Table 5: MMLU 5-shot accuracy comparison. The rows with the lowest (blue) and highest (red) average scores for each method are highlighted.

| Method | Setting | Accuracy (%) | | | | |
|---|---|---|---|---|---|---|
| | | Hums. | STEM | Social. | Other | Avg. |
| Baseline | - | 43.6 | 37.1 | 51.4 | 53.0 | 45.9 |
| Pruning to Zero | Seed #1 | 34.0 | 31.8 | 38.5 | 42.8 | 36.4 |
| | Seed #2 | 39.9 | 35.9 | 48.3 | 49.9 | 43.0 |
| | Seed #3 | 39.5 | 38.3 | 47.4 | 49.1 | 43.1 |
| | Seed #4 | 40.4 | 37.5 | 49.6 | 50.6 | 44.0 |
| Fine-tuning | Seed #1 | 50.2 | 44.1 | 63.2 | 60.4 | 54.0 |
| | Seed #2 | 51.3 | 43.2 | 62.6 | 60.9 | 54.1 |
| | Seed #3 | 51.2 | 42.6 | 62.1 | 60.5 | 53.7 |
| | Seed #4 | 51.2 | 44.3 | 62.8 | 60.6 | 54.3 |

To determine whether the initial weight sensitivity of a model to a downstream task could predict its final fine-tuning performance, we used the performance degradation from our "Pruning to Zero" experiment as a proxy for sensitivity (Table 5). The results revealed a high variance in sensitivity depending on the random seed; for instance, the accuracy drop from the baseline ranged from a minimal 1.9 % (Seed #4 from the previous table) to a substantial 9.5 % (Seed #1), indicating that different sets of weights with varying importance were pruned.

Notably, we observed no correlation between this initial weight sensitivity and the final fine-tuning accuracy. Despite the wide disparity in the pruning outcomes, all fine-tuning runs converged to a narrow and high-performance bracket (e.g., 53.7 % to 54.3 %). This suggests that the fine-tuning process is highly robust and capable of overriding the initial weight state of the model to find a consistent solution.

## B    GRADIENT ANALYSIS OF INDEX ALIGNMENT

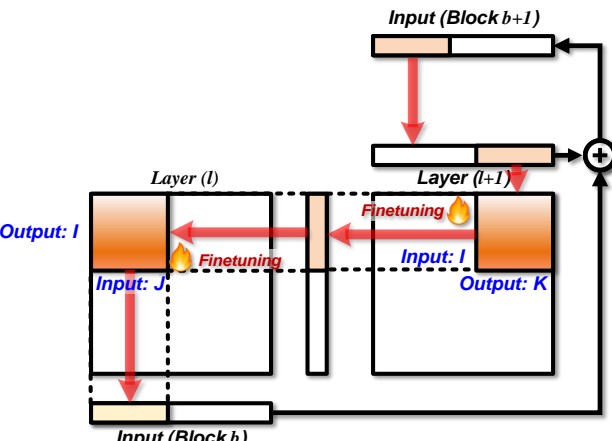

Figure 6: Example of disrupted gradient flow in the absence of index alignment

In this section, we analytically demonstrate the benefit of index alignment in RAPA through a formal mathematical formulation. As defined in Section 3.2.2, we consider a simplified structure in which a single block consists of two consecutive linear layers connected via a residual connection. Let $I, J, K, T \subset \{1, \ldots, d\}$ denote the selected index subsets over the hidden dimension $d$, where each subset corresponds to the input/output subspaces involved in fine-tuning.

Mathematically, the forward propagation through a block without applying any alignment can be defined as:

$$x_I^{(l+1)} = W_{I,J}^{(l)} x_J^{(l)} \tag{8}$$

$$x_K^{(l+2)} = W_{K,T}^{(l+1)} x_T^{(l+1)} + x_K^{(l)} \tag{9}$$

### B.1 LAYER-WISE ALIGNMENT: ALIGNING OUTPUT-INPUT INDICES ACROSS LAYERS

In layer-wise alignment, the output indices of a layer are matched with the input indices of the next layer to enhance gradient propagation. Based on the forward propagation defined in Eqs. 8 and 9, the loss gradient with respect to the trainable weight $W_{K,T}^{(l+1)}$ at layer $l+1$ is given by

$$\frac{\partial \mathcal{L}}{\partial W_{K,T}^{(l+1)}} = \frac{\partial \mathcal{L}}{\partial x_K^{(l+2)}} \frac{\partial x_K^{(l+2)}}{\partial W_{K,T}^{(l+1)}} \tag{10}$$

The weights corresponding to the selected subspace of the layer $l+1$ are updated during back-propagation as follows:

$$W_{K,T}^{(l+1)} \leftarrow W_{K,T}^{(l+1)} - \eta \frac{\partial \mathcal{L}}{\partial W_{K,T}^{(l+1)}} \tag{11}$$

where $\eta$ is the learning rate. By applying the chain rule, we can similarly compute the gradient of the selected subspace weights at layer $l$, $W_{I,J}^{(l)}$ as

$$\frac{\partial \mathcal{L}}{\partial W_{I,J}^{(l)}} = \frac{\partial \mathcal{L}}{\partial x_K^{(l+2)}} \frac{\partial x_K^{(l+2)}}{\partial x_I^{(l+1)}} \frac{\partial x_I^{(l+1)}}{\partial W_{I,J}^{(l)}} \tag{12}$$

where $\frac{\partial x_K^{(l+2)}}{\partial x_I^{(l+1)}} = W_{K,I}^{(l+1)}$ (note that $W_{K,I}^{(l+1)}$ is not included in the selected trainable subspace), and $\frac{\partial x_I^{(l+1)}}{\partial W_{I,J}^{(l)}} = x_J^{(l)}$. Accordingly, the gradient becomes:

$$\frac{\partial \mathcal{L}}{\partial W_{I,J}^{(l)}} = \frac{\partial \mathcal{L}}{\partial x_K^{(l+2)}} \boldsymbol{W_{K,I}^{(l+1)} x_J^{(l)}} \tag{13}$$

The gradient in Eq. 13 reveals that the update of $W_{I,J}^{(l)}$ is modulated by the factor $W_{K,I}^{(l+1)}$ from the subsequent layer. Since Eq. 9 assumes that only the weights indexed by $(K, T)$ are included in the *trainable subspace* of layer $l+1$, the element $W_{K,I}^{(l+1)}$ remains *frozen* during the update step of layer $l$ when $T \neq I$. Consequently, the back-propagated gradient that reaches $W_{I,J}^{(l)}$ is scaled by this fixed weight.

When the index sets are **aligned** ($T = I$), the factor $W_{K,I}^{(l+1)}$ is updated at every iteration, allowing the gradient to flow more directly into $W_{I,J}^{(l)}$ and accelerating convergence, as discussed in Section 3.2.2.

### B.2 BLOCK-WISE ALIGNMENT: ALIGNING WITHIN RESIDUAL-CONNECTED LAYERS

The residual connection within the block transmits both activations and gradients directly across layers. Such residual paths preserve gradient magnitude and stabilize the training of deep neural networks (He et al., 2016b).

**Gradient decomposition** For clarity, we focus on a single residual block, referred to as the *target block*, whose final layer index is $l+2$. The gradient arriving at the output of the target block during back-propagation can be decomposed into two additive parts:

$$\frac{\partial \mathcal{L}}{\partial x_K^{(l+2)}} = g_K{}^{linear} + g_K{}^{residual} \tag{14}$$

where $g_K{}^{linear}$ is the gradient that has passed through all layers after the target block, and $g_K{}^{residual}$ flows along the residual connection.

**Why align indices inside the block?** As defined in Appendix B.1, the layer-wise alignment method aligns index sets between successive layers ($T = I$).

In this section, to motivate the need for block-wise alignment, we consider a block with a residual connection from input to output. We denote the input index set as $J$ and the output index set as $K$. When the output index subspace $K$ and input index subspace $J$ of the block are not aligned, the back-propagation gradient expands as follows:

$$\frac{\partial \mathcal{L}}{\partial x_J^{(l)}} = \frac{\partial \mathcal{L}}{\partial x_K^{(l+2)}} \frac{\partial x_K^{(l+2)}}{\partial x_I^{(l+1)}} \frac{\partial x_I^{(l+1)}}{\partial x_J^{(l)}} \tag{15}$$

$$= \frac{\partial \mathcal{L}}{\partial x_K^{(l+2)}} \left( W_{K,I}^{(l+1)} W_{I,J}^{(l)} + \delta_r; J, K \right), \quad \delta_{r;J,K} = \begin{cases} 1 & \text{if } r \in J \cap K \\ 0 & \text{if } r \notin J \cap K \end{cases} \tag{16}$$

where $\delta_{r;J,K}$ is the indicator function. If we align the output and shortcut index sets such that $K = J$, the expression simplifies to:

$$\frac{\partial \mathcal{L}}{\partial x_J^{(l)}} = \frac{\partial \mathcal{L}}{\partial x_J^{(l+2)}} \left( W_{J,I}^{(l+1)} W_{I,J}^{(l)} + \mathbf{1} \right) \tag{17}$$

The additive identity term arises only when the same index set is used on both sides of the residual connection ($K = J$). This identity term creates a direct gradient path from the output of the block back to its input layer ($l$), as formalized in Eq. 14, effectively shortening the gradient route.

As a result, it helps preserve gradient magnitude and reduces training loss by approximately 3% (see Figure 4 in Section 3.2.2).

## B.3 GLOBAL ALIGNMENT: INDEX ALIGNMENT ACROSS RESIDUAL BLOCKS

The benefits of residual connections extend not only within a block but also across multiple sequential blocks. To fully exploit this, we align indices between adjacent blocks. Global alignment extends the index-selection rule of RAPA so that the same index set $T$ is used for fine-tuning in all consecutive blocks.

For forward propagation of block $b$ (refer to Eq. 3 in Section 3.2.2 for notation), we write

$$x_J^{(b+1)} = x_J^{(b)} + \mathcal{F}\left( W_{J,J}^{(b)}, x_J^{(b)} \right) \tag{18}$$

where $\mathcal{F}(\cdot)$ is the two-layer transformation inside the block. Repeating this for $m$ successive blocks yields

$$x_J^{(b+m)} = x_J^{(b)} + \sum_{i=0}^{m-1} \mathcal{F}\left( W_{J,J}^{(b+i)}, x_J^{(b+i)} \right). \tag{19}$$

Applying the chain rule, the gradient back-propagated to the initial input is represented by

$$\frac{\partial \mathcal{L}}{\partial x_J^{(b)}} = \underbrace{\frac{\partial \mathcal{L}}{\partial x_J^{(b+m)}}}_{\text{direct shortcut}} \left[ 1 + \underbrace{\frac{\partial}{\partial x_J^{(b)}} \sum_{i=0}^{m-1} \mathcal{F}\left( W_{J,J}^{(b+i)}, x_J^{(b+i)} \right)}_{\text{via learned weights}} \right]. \tag{20}$$

As in block-wise alignment, the term "1" directly links the final loss back to the shallow block without traversing intermediate weights. This direct gradient path:

- Shortens the effective back-propagation path length.
- Strengthens early-layer updates.
- Mitigates vanishing gradients even across many blocks.

## C ATTENTION-LAYER INDEX ALIGNMENT

In this section, we clarify (i) which attention parameters are selected by RAPA, (ii) why hidden-dimension index alignment still matters after forming the attention scores $S = QK^T/\sqrt{d_h}$, and (iii) how this interacts with the multi-head structure and the low-rank nature of $QK^T$.

### C.1 WHICH ATTENTION PARAMETERS ARE SELECTED?

In each multi-head attention block, we write the projections as

$$Q = xW_Q, \quad K = xW_K, \quad V = xW_V, \quad z = AV, \quad y = x + zW_O,$$

where $x \in \mathbb{R}^{d_{\text{model}}}$ is the residual stream and $A$ is the attention weight matrix.

The core RAPA update always includes $W_V$ and $W_O$, using the same input/output index sets as the residual stream and FFN layers. As an extension, we also align the index sets of $W_Q$ and $W_K$ with those of $W_V/W_O$. This additional Q/K alignment is not required for RAPA to function, but it further improves gradient flow, as shown by the ablation in Section C.4.

### C.2 WHY HIDDEN INDEX ALIGNMENT STILL MATTERS

After forming

$$S = \frac{QK^T}{\sqrt{d_h}} \in \mathbb{R}^{T \times T},$$

the hidden dimension is no longer explicitly visible: each entry $S_{ij}$ is a scalar score between a query token $i$ and a key token $j$. However, each score is still a bilinear function of the hidden coordinates:

$$S_{ij} = \frac{1}{\sqrt{d_h}}\langle q_i, k_j \rangle = \frac{1}{\sqrt{d_h}}\sum_{\ell=1}^{d_h} q_i[\ell]\, k_j[\ell].$$

RAPA's index alignment acts *before* this contraction over the head dimension. Let $I \subset \{1, \ldots, d_h\}$ be the aligned index set shared by $W_Q, W_K, W_V, W_O$ and the FFN blocks, and decompose

$$q_i = (q_i^I,\ q_i^{\bar{I}}), \quad k_j = (k_j^I,\ k_j^{\bar{I}}).$$

RAPA only trains the slices of $W_Q$ and $W_K$ that produce $q_i^I$ and $k_j^I$; the remaining coordinates $q_i^{\bar{I}}, k_j^{\bar{I}}$ stay essentially frozen. For a small update, the change in the score is then

$$\Delta S_{ij} = \frac{1}{\sqrt{d_h}}\Big(\langle \Delta q_i^I,\ k_j^I \rangle + \langle q_i^I,\ \Delta k_j^I \rangle\Big),$$

because the non-aligned coordinates $\bar{I}$ do not move.

In other words: (i) only the aligned hidden coordinates $I$ contribute to the *trainable* part of the score change $\Delta S$, and (ii) those same coordinates $I$ are exactly where $W_V, W_O$, and the FFN are updated. Thus, even though $S$ lives in token–token space, the directions along which we are allowed to change $S$ are controlled by which hidden indices we make trainable in $W_Q$ and $W_K$. Aligning Q/K indices with V/O/FFN indices ensures that changes in the attention scores are produced by the same hidden channels that later carry the value and residual updates.

### C.3 GRADIENT VIEW AND PROJECTION

The above intuition is reflected in a gradient-based view. For a single head, with

$$S = \frac{QK^T}{\sqrt{d_h}}, \quad A = \text{softmax}(S), \quad z = AV, \quad y = x + zW_O,$$

a small change in $W_Q$ leads to

$$\Delta \mathcal{L}_Q = \frac{1}{\sqrt{d_h}}\langle G_Q, \Delta W_Q \rangle,$$

where the effective gradient on $W_Q$ factors as

$$G_Q = x^T \mathcal{J}_{\text{sm}}(S)^T G(VW_O)^T K.$$

Here $G = \partial\mathcal{L}/\partial y$ and $\mathcal{J}_{\text{sm}}(S)$ is the Jacobian of the row-wise softmax.

RAPA restricts updates to a projected subspace,

$$\Delta W_Q \leftarrow P_{\text{in}}^{QK} \Delta W_Q \, P_{\text{out}}^{QK},$$

so the useful loss decrease is governed by

$$-\Delta\mathcal{L}_Q \propto \left\| P_{\text{in}}^{QK} G_Q P_{\text{out}}^{QK} \right\|.$$

Choosing $P_{\text{in}}^{QK}$ and $P_{\text{out}}^{QK}$ to share the same indices as the projections used for $W_V$, $W_O$, and the FFN means that the gradient directions that matter for Q/K and for V/O/FFN coincide. This is precisely the pathway alignment principle extended from FFNs to attention.

## C.4 EMPIRICAL EFFECT OF Q/K ALIGNMENT

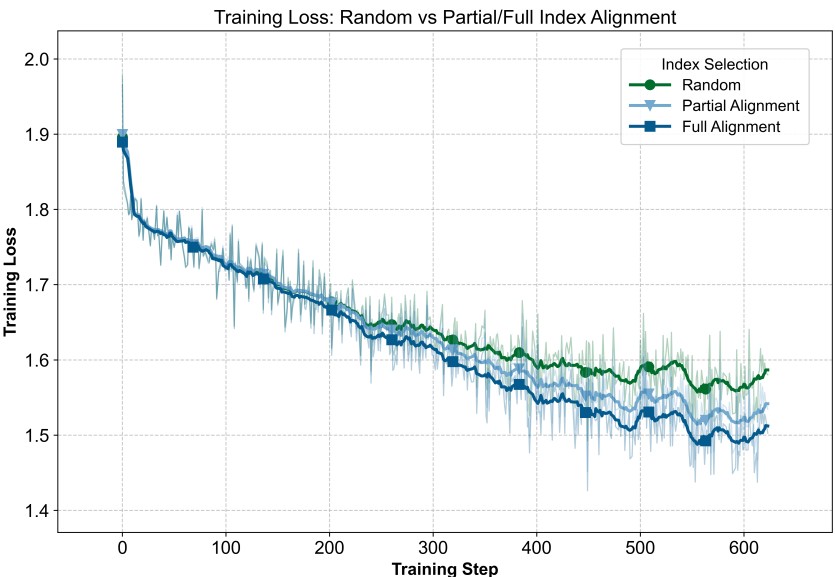

Figure 7: Training loss for LLaMA2-7B under three index-selection schemes. Random uses random indices. Partial Alignment uses aligned indices for V/O and FFN only. Full Alignment uses aligned indices for V/O, FFN, and Q/K. Thin curves show raw per-step training loss, and thick curves show a moving-average smoothed loss over steps.

We now make the empirical role of Q/K alignment explicit via an ablation on LLaMA2-7B (5-shot MMLU). We compare: (i) a variant that aligns only V/O and FFN indices, and (ii) a variant that additionally aligns Q/K indices.

Table 6: Ablation on attention Q/K alignment for LLaMA2-7B (5-shot MMLU). PPL is training loss perplexity (lower is better).

| Method | PPL↓ | Human. | STEM | Social. | Other | Average |
|---|---|---|---|---|---|---|
| Partial alignment (V/O + FFN only) | 4.62 | 49.1 | 42.1 | 60.4 | 59.0 | 52.2 |
| Full alignment (including Q/K) | 4.49 | 50.0 | 42.4 | 60.8 | 59.1 | 52.7 |

On LLaMA2-7B, aligning only V/O and FFN indices yields 52.2% average MMLU accuracy, while additionally aligning Q/K indices increases this to 52.7% (+0.5 points) with a slightly lower perplexity (Table 6). Figure 7 shows the corresponding training-loss trajectories for the three index-selection schemes. As suggested by the analysis above, Q/K alignment has a smaller effect than aligning FFN and V/O (since Q/K influence the loss more indirectly through attention scores), but it still provides consistent gains on top of the core RAPA pathway alignment.

# D    EXTENSION OF RAPA TO MOE RESIDUAL BLOCKS

In this section, we show that the *index alignment* mechanism introduced for dense Transformer architectures extends naturally to MoE architectures as well. By re-writing MoE residual blocks in the aligned subspace and applying the same residual-path analysis, we obtain an analogous "identity + aligned update" structure for expert FFNs, with the router acting only as a smooth reweighting factor. The derivations below make this correspondence explicit.

## D.1    MOE RESIDUAL BLOCKS STILL ADMIT THE SAME PATHWAY STRUCTURE.

Consider a standard MoE FFN block at layer $b$:

$$x^{(b+1)} = x^{(b)} + \sum_{e=1}^{E} p_e^{(b)}\big(x^{(b)}\big)\, F_e^{(b)}\big(W^{(b,e)}, x^{(b)}\big),$$

where $p_e^{(b)}(x^{(b)})$ are router probabilities and $F_e^{(b)}$ is the expert FFN.

In the MoE version of RAPA, we choose a common index set $J \subset \{1,\ldots,d\}$ with $|J| = k$, and restrict *all experts in this block* to update only the coordinates in $J$. That is, each expert FFN operates on $x_J^{(b)}$ and uses weights $W_{J,J}^{(b,e)}$. Focusing on the $J$-subspace, the block becomes

$$x_J^{(b+1)} = x_J^{(b)} + \sum_{e=1}^{E} p_e^{(b)}\big(x^{(b)}\big)\, F_e^{(b)}\big(W_{J,J}^{(b,e)}, x_J^{(b)}\big) =: x_J^{(b)} + \tilde{F}^{(b)}\big(\tilde{W}^{(b)}, x_J^{(b)}\big).$$

If we stack $m$ such MoE blocks and apply the chain rule, we obtain

$$\frac{\partial \mathcal{L}}{\partial x_J^{(b)}} = \frac{\partial \mathcal{L}}{\partial x_J^{(b+m)}}\left[I + \frac{\partial}{\partial x_J^{(b+m)}} \sum_{t=0}^{m-1} \tilde{F}^{(b+t)}\big(\tilde{W}^{(b+t)}, x_J^{(b+t)}\big)\right].$$

Crucially, as in the dense case, there is an *identity term on $x_J^{(b)}$* plus additional terms from the experts. Thus, even with MoE, aligning the same index set $J$ across MoE blocks preserves the *direct residual shortcut* in the $J$-subspace and the same "pathway" structure that our dense analysis relies on.

## D.2    LAYER-WISE INDEX ALIGNMENT INSIDE MOE EXPERTS.

For an expert $e$, we write the FFN as

$$f_e(x) = W_2^{(b,e)}\, \phi\big(W_1^{(b,e)} x\big).$$

To apply layer-wise alignment, we choose indices $(I, J)$ such that

- $W_1^{(b,e)}$ is trainable only on the block $(I, J)$, and
- $W_2^{(b,e)}$ is trainable only on the block $(J, I)$,

with the same $(I, J)$ shared across all experts in that layer. Restricting to $x_J$, the MoE FFN contribution to the residual becomes

$$x_J^{(b+1)} = x_J^{(b)} + \sum_{e=1}^{E} p_e^{(b)}\big(x^{(b)}\big)\, W_{2,J,I}^{(b,e)}\, \phi\big(W_{1,I,J}^{(b,e)} x_J^{(b)}\big).$$

For an expert's first linear layer, the gradient takes the form

$$\frac{\partial \mathcal{L}}{\partial W_{1,I,J}^{(b,e)}} = \frac{\partial \mathcal{L}}{\partial x_J^{(b+1)}} \cdot p_e^{(b)}\big(x^{(b)}\big) \cdot \frac{\partial f_e}{\partial W_{1,I,J}^{(b,e)}}.$$

When we expand $\partial f_e/\partial W_{1,I,J}^{(b,e)}$, we obtain factors of the form

$$W_{2,J,I}^{(b,e)}\, x_J^{(b)},$$

which is exactly analogous to the dense case, up to the multiplicative router weight $p_e^{(b)}(x^{(b)})$. Thus, layer-wise index alignment within experts preserves the same linear "pathway" coupling between the input $x_J^{(b)}$ and the output residual $x_J^{(b+1)}$; the MoE router just reweights each expert's contribution.

## D.3 ROUTER GRADIENTS AND STABILITY.

If we include the router gradient, the derivative of $\tilde{F}^{(b)}$ with respect to $x_J^{(b)}$ contains both

- terms where $p_e^{(b)}$ acts as a scale factor on expert gradients, and
- additional terms of the form

$$\sum_{e=1}^{E} \frac{\partial p_e^{(b)}\big(x^{(b)}\big)}{\partial x_J^{(b)}} F_e^{(b)}\big(W_{J,J}^{(b,e)}, x_J^{(b)}\big),$$

which are still defined inside the same $J$-subspace.

These router-related terms add extra contributions but do not remove the identity shortcut on $x_J^{(b)}$, and they do not require a different alignment rule. In our experiments, we keep the router parameters' training scheme identical to the MoE baseline (no special RAPA constraint on router weights), and we did not observe any instability attributable to RAPA. Hence, the core residual alignment argument (identity + aligned MoE updates in $J$) continues to hold, while the router acts as a smooth, subspace-preserving reweighting mechanism.

## D.4 PRELIMINARY MoE EXPERIMENTS.

To check that this analysis is not purely theoretical, we also conducted preliminary experiments on a MoE LLM (DeepSeek-MoE-16B). Under the same parameter budget, applying our index alignment to the MoE architecture—enforcing a common aligned index set across layers and across experts—leads to (i) lower training perplexity (computed as $\exp$ of the final training loss) compared to random index selection, and (ii) a small but consistent improvement in downstream metrics (MMLU accuracy). The quantitative results are summarized in Table 7, and the corresponding training-loss trajectories across seeds are shown in Figure 8.

Table 7: Preliminary MoE results on DeepSeek-MoE-16B. PPL is training perplexity, computed as $\exp$ of the final training loss; other columns report 5-shot MMLU accuracies (%).

| Method | Train PPL ↓ | Human. | STEM | Social. | Other | Average |
|---|---|---|---|---|---|---|
| Baseline | 8.71 | 41.3 | 36.9 | 51.2 | 51.7 | 44.8 |
| PaCA (Square, Random) | 4.34 | 47.8 | 40.8 | 57.9 | 57.3 | 50.5 |
| RAPA (Square, Aligned) | 3.74 | 48.9 | 42.1 | 59.1 | 59.5 | 51.9 |

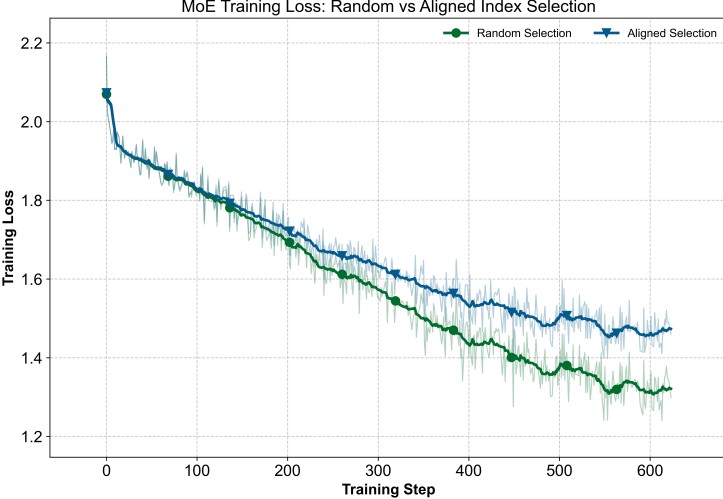

Figure 8: Training loss trajectories for the MoE model (DeepSeek-MoE-16B) under random index selection and proposed aligned index selection.

# E    DETAILED TRAINING LOSS CURVES

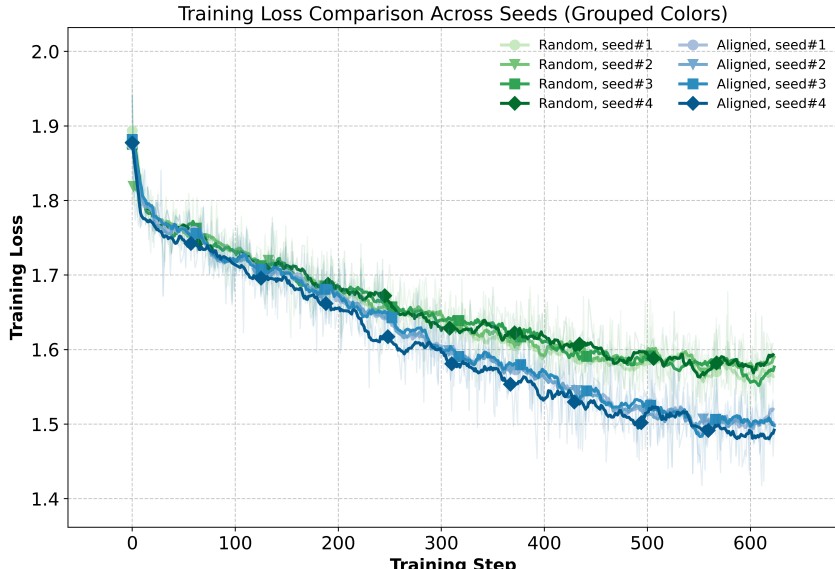

Figure 9: Detailed training loss curves for aligned vs. random index selection across four different random seeds. The grouped colors (blue hues for aligned, green hues for random) and unique markers identify each individual run. This demonstrates that the faster convergence of the aligned method is consistent and robust across various initializations.

# F    5-SHOT MMLU ACCURACY COMPARISONS UNDER VARIOUS RANKS

Table 8: 5-shot accuracy (%) on MMLU dataset by rank and method.

| Method | Rank | # Params(M) | Accuracy (%) | | | | |
|---|---|---|---|---|---|---|---|
| | | | Hums. | STEM | Social. | Other | Avg. |
| Baseline | - | - | 43.6 | 37.1 | 51.4 | 53.0 | 45.9 |
| LoRA | N/A | - | - | - | - | - | - |
| DoRA | N/A | - | - | - | - | - | - |
| MosLoRA | N/A | - | - | - | - | - | - |
| PaCA | 1 | 1.4M | 44.6 | 38.1 | 54.0 | 53.8 | 47.2 |
| RAPA (Ours) | 60 | 1.4M | **46.7** | **40.6** | **56.3** | **56.4** | **49.6** |
| LoRA | 1 | 2.5M | 46.7 | **42.6** | 57.9 | 57.6 | 50.6 |
| DoRA | 1 | 3.9M | 47.5 | 41.8 | 58.8 | 58.0 | 51.0 |
| MosLoRA | 1 | 2.5M | **48.2** | 40.9 | **59.8** | 58.6 | **51.4** |
| PaCA | 2 | 2.7M | 46.5 | 39.8 | 55.7 | 54.7 | 48.8 |
| RAPA (Ours) | 80 | 2.5M | 47.6 | 41.8 | 58.8 | **58.4** | 51.1 |
| LoRA | 8 | 20.0M | 48.5 | 41.2 | 57.3 | 56.5 | 50.6 |
| DoRA | 8 | 21.4M | 48.7 | 42.3 | 58.3 | 57.6 | 51.3 |
| MosLoRA | 8 | 20.0M | 46.6 | 42.2 | **60.8** | 57.4 | 51.1 |
| PaCA | 16 | 21.8M | 48.7 | 41.7 | 58.7 | 57.6 | 51.2 |
| RAPA (Ours) | 228 | 20.1M | **50.1** | **42.6** | **60.8** | **59.1** | **52.8** |
| LoRA | 64 | 159.9M | **51.2** | 43.1 | 61.7 | 60.1 | 53.6 |
| DoRA | 64 | 161.3M | 51.0 | 43.6 | 61.8 | 60.1 | 53.7 |
| MosLoRA | 64 | 160.8M | 50.4 | 42.8 | 61.5 | 59.5 | 53.1 |
| PaCA | 128 | 174.1M | 50.7 | 43.3 | **63.5** | 60.1 | 53.9 |
| RAPA (Ours) | 644 | 160.1M | **51.2** | **44.3** | 62.8 | **60.6** | **54.3** |

# G    IMPLEMENTATION DETAILS

Algorithm 1 instantiates *all three* alignment rules in a single pass over the residual blocks.

---
**Algorithm 1** Index Alignment of Trainable Weights

---
1: **Input:** residual blocks $(\mathcal{B}_1, \ldots, \mathcal{B}_L)$, hidden size $d$, square side $k$, seed $s$
2: $I \leftarrow$ RANDOMPICK$(d, k, s)$
3: **for** $b = 1$ **to** $L$ **do**
4:     $\mathcal{B}_b.\texttt{input} \leftarrow I$                                               // block-wise and global alignment
5:     $\texttt{prevOut} \leftarrow I$
6:     **for all** linear layer $\ell$ **in** $\mathcal{B}_b$ **do**
7:         $\ell.\texttt{input} \leftarrow \texttt{prevOut}$                                     // layer-wise alignment
8:         $\ell.\texttt{output} \leftarrow$ RANDOMPICK$(d, k)$
9:         $\texttt{prevOut} \leftarrow \ell.\texttt{output}$
10:     **end for**
11:     $\mathcal{B}_b.\texttt{output} \leftarrow I$                                           // last layer output of a block
12: **end for**
13: **return** mapping $\{\ell.\texttt{input}, \ell.\texttt{output}\}$

---

# H    EXPERIMENTAL DETAILS

Table 9: Hyperparameters for the row/column weight ratio sweep experiment on LLaMA2-7B.

| Fixed Hyperparameters | |
|---|---|
| Method | RAPA |
| Training Precision | 16-bit mixed precision |
| Total Trainable Weights | 4,096 per layer |
| Optimizer | AdamW |
| LR Scheduler | Cosine |
| Batch Size | 8 |
| Sequence Length | 512 |
| Epochs | 1 |
| Target Modules | Q, K, V, O layers in Attention |

**Sweep Configuration**

Rank (column) is swept through powers of two from 1 to 4096. Rank (row) is set such that Rank (row) $\times$ Rank (column) = 4096. Below are representative examples.

| Rank (column) | Rank (row) | Ratio (column:row) | alpha |
|---|---|---|---|
| 1 | 4096 | 100/0 | 4 |
| 2 | 2048 | 92/8 | 8 |
| $\vdots$ | | | |
| 64 | 64 | 50/50 | 64 |
| $\vdots$ | | | |
| 2048 | 2 | 8/92 | 8192 |
| 4096 | 1 | 0/100 | 16384 |

Table 10: Hyperparameters for fine-tuning LLaMA2-7B and LLaMA2-13B on MMLU using PEFT (parentheses indicate settings for LLaMA2-13B).

| Hyperparameters | LoRA | DoRA | MosLoRA | PaCA | RAPA |
|---|---|---|---|---|---|
| Training Precision | 16-bit mixed precision (Micikevicius et al., 2017) | | | | |
| Rank | 128 | 128 | 128 | 256 | 910 (1018) |
| $\alpha$ | 64 | 64 | 64 | 128 | 455 (509) |
| DropOut | 0.1 | 0.1 | 0.1 | - | - |
| Optimizer | AdamW (Loshchilov & Hutter, 2017) | | | | |
| LR | 1e-4, 2e-4, 3e-4 | | | | |
| LR Scheduler | cosine | | | | |
| Batch Size | 8 (4) | | | | |
| Gradient Accumulation Steps | 1 (2) | | | | |
| Sequence Length | 512 | | | | |
| Warmup Steps | 100 | | | | |
| Epochs | 1 | | | | |
| Target Modules | Q, K, V, O, Up, Down, Gate | | | | |

Table 11: Hyperparameters for fine-tuning LLaMA2-7B on Commonsense Reasoning using PEFT.

| Hyperparameters | LoRA | DoRA | MosLoRA | PaCA | RAPA |
|---|---|---|---|---|---|
| Training Precision | 16-bit mixed precision (Micikevicius et al., 2017) | | | | |
| Rank | 128 | 128 | 128 | 256 | 910 |
| $\alpha$ | 64 | 64 | 64 | 128 | 455 |
| DropOut | 0.1 | 0.1 | 0.1 | - | - |
| Optimizer | AdamW (Loshchilov & Hutter, 2017) | | | | |
| LR | 2e-4, 3e-4 | | | | |
| LR Scheduler | cosine | | | | |
| Batch Size | 8 | | | | |
| Gradient Accumulation Steps | 1 | | | | |
| Sequence Length | 512 | | | | |
| Warmup Steps | 100 | | | | |
| Epochs | 1 | | | | |
| Target Modules | Q, K, V, O, Up, Down, Gate | | | | |

Table 12: Hyperparameters for instruction tuning Mistral-7B-v0.1 on OASST1 dataset using PEFT.

| Hyperparameters | LoRA | DoRA | MosLoRA | PaCA | RAPA |
|---|---|---|---|---|---|
| Training Precision | 16-bit mixed precision (Micikevicius et al., 2017) | | | | |
| Rank | 64 | 64 | 64 | 128 | 602 |
| $\alpha$ | 1 | | | | |
| DropOut | 0.1 | 0.1 | 0.1 | - | - |
| Optimizer | AdamW (Loshchilov & Hutter, 2017) | | | | |
| LR | 1e-3, 5e-4 | | | | |
| LR Scheduler | linear | | | | |
| Batch Size | 4 | | | | |
| Gradient Accumulation Steps | 4 | | | | |
| Sequence Length | 1024 | | | | |
| Warmup Steps | 61 | | | | |
| Epochs | 1 | | | | |
| Target Modules | Q, K, V, O, Up, Down, Gate | | | | |

# I  OVERHEAD ABLATION

To verify that the overhead pattern observed in the main text is not specific to a particular model width, we repeat our profiling on two decoder-only Transformer models with different hidden dimensions: a smaller model with $h_{\text{dim}} = 1024$ and a wider model with $h_{\text{dim}} = 8192$. For each model, we compare a frozen baseline, an adapter-based (AB) method, and an adapter-free (AF) counterpart under the same benchmarking protocol as in our main-text overhead analysis.

Figures 10(a)–(d) report the absolute number of GEMM kernel launches and the corresponding forward-time overhead for the two widths. In both regimes, the AB method consistently triggers approximately $3\times$ more GEMM kernels than the baseline, which translates into about $35$–$45\%$ forward-time overhead. By contrast, the AF method closely tracks the baseline in both kernel count and runtime, incurring only a few percent overhead.

Figures 10(e)–(h) further break down the kernel-count and matrix-multiplication time *ratios* across a range of batch sizes, sequence lengths, and adapter ranks $r$. The overhead of AB is largely insensitive to $r$ and $h_{\text{dim}}$, but grows with batch size and sequence length due to the structural cost of executing additional small kernels. In all settings, AF remains near parity with the baseline. These results confirm that the structural inefficiency of adapter modules is robust across model width, and that our adapter-free design can retain high efficiency even for wider models.

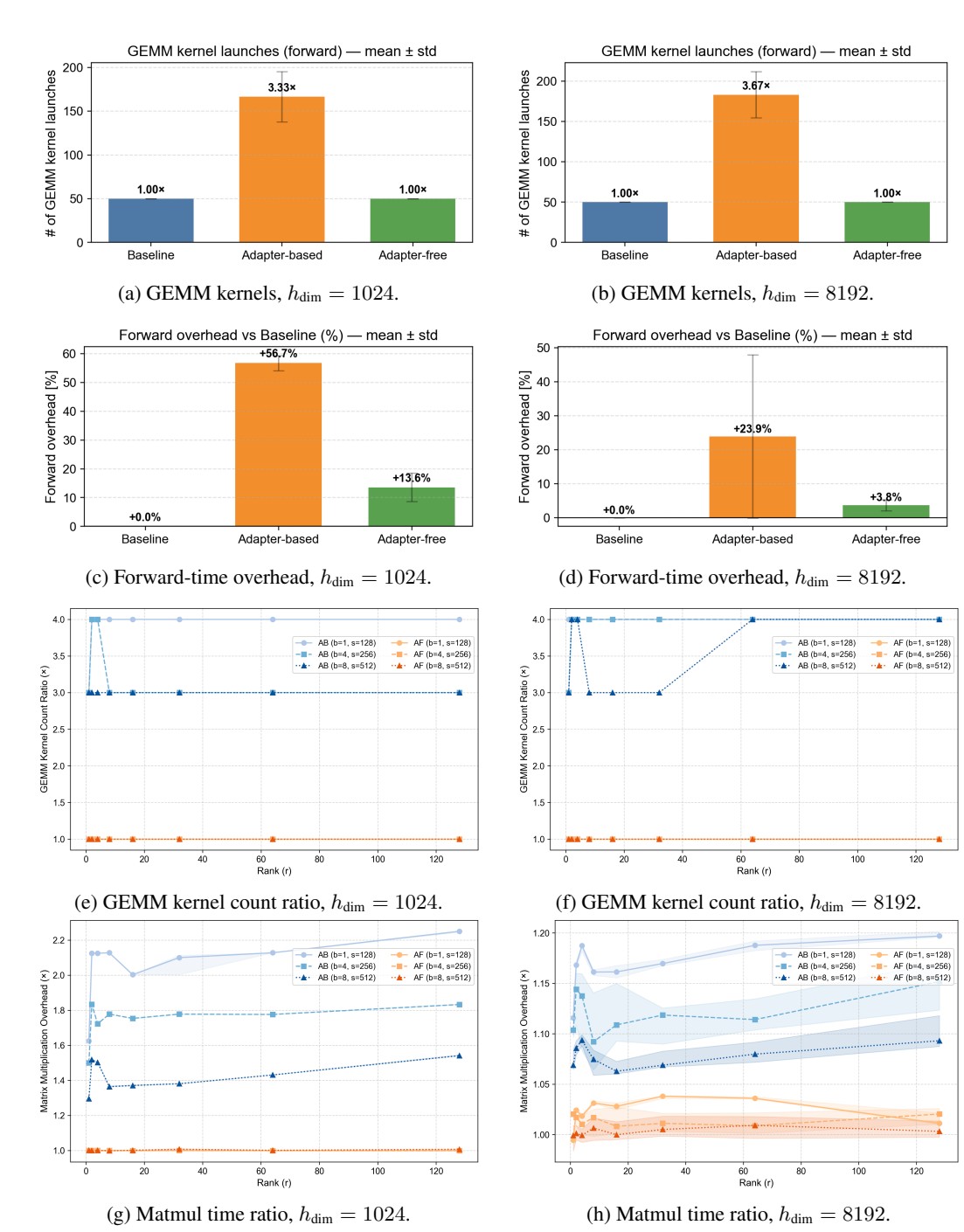

(a) GEMM kernels, $h_{\text{dim}} = 1024$.

(b) GEMM kernels, $h_{\text{dim}} = 8192$.

(c) Forward-time overhead, $h_{\text{dim}} = 1024$.

(d) Forward-time overhead, $h_{\text{dim}} = 8192$.

(e) GEMM kernel count ratio, $h_{\text{dim}} = 1024$.

(f) GEMM kernel count ratio, $h_{\text{dim}} = 8192$.

(g) Matmul time ratio, $h_{\text{dim}} = 1024$.

(h) Matmul time ratio, $h_{\text{dim}} = 8192$.

Figure 10: Overhead ablation for adapter-based (AB) and adapter-free (AF) methods on two model widths. (a, b) GEMM kernel counts; (c, d) forward-time overhead; (e, f) kernel-count ratios; (g, h) matmul-time ratios, all relative to the frozen baseline.

## J   RECURSIVE GRADIENT VARIANCE BOUND: ALIGNED VS. RANDOM SELECTION

### J.1   SETUP

Let $g_b \in \mathbb{R}^{d_b}$ denote the backpropagated gradient entering the $b$-th block, and let $J_b \in \mathbb{R}^{d_{b-1} \times d_b}$ be its local Jacobian, which linearly maps perturbations from layer $b$ to layer $b-1$:

$$g_{b-1} = J_b\, g_b + \xi_b.$$

Here, $\xi_b$ is a zero-mean stochastic term

$$\mathbb{E}[\xi_b \mid \mathcal{F}_b] = 0, \qquad \mathbb{E}\|\xi_b\|_2^2 \le \sigma_b^2,$$

modeling the variance injection from mini-batch sampling, normalization noise, or quantization.

Each block updates only a $k$-dimensional subspace of its input/output through orthogonal projections

$$P_{S_b^{(\mathrm{in})}} \in \mathbb{R}^{d_b \times d_b}, \qquad P_{S_b^{(\mathrm{out})}} \in \mathbb{R}^{d_{b-1} \times d_{b-1}},$$

where $S_b^{(\mathrm{in})}$ and $S_b^{(\mathrm{out})}$ are the index sets of selected input and output dimensions, respectively. Thus, the aligned backpropagation can be expressed as

$$g_{b-1} = P_{S_{b-1}^{(\mathrm{out})}} \big( J_b P_{S_b^{(\mathrm{in})}} g_b + \xi_b \big).$$

### J.2   VARIANCE RECURSION

Taking the conditional variance and using the submultiplicativity of operator norms gives

$$\mathrm{Var}(g_{b-1}) \;\le\; \big\| P_{S_{b-1}^{(\mathrm{out})}} J_b P_{S_b^{(\mathrm{in})}} \big\|_2^2 \, \mathrm{Var}(g_b) + \mathbb{E} \big\| P_{S_{b-1}^{(\mathrm{out})}} \xi_b \big\|_2^2.$$

We define the *effective spectral norm* of block $b$ as

$$\rho_b := \big\| P_{S_{b-1}^{(\mathrm{out})}} J_b P_{S_b^{(\mathrm{in})}} \big\|_2,$$

which measures the local gradient amplification of block $b$.

Unrolling the recursion over $m$ blocks yields the product–sum form:

$$\mathrm{Var}(g_0) \le \Big( \prod_{b=1}^{m} \rho_b^2 \Big) \mathrm{Var}(g_m) + \sum_{i=1}^{m} \Big( \prod_{b=1}^{i-1} \rho_b^2 \Big) \mathbb{E} \big\| P_{S_{i-1}^{(\mathrm{out})}} \xi_i \big\|_2^2.$$

The first term represents the variance propagated from the top layer ($g_m$), and the summation term represents the cumulative variance injected by intermediate layers.

### J.3   EFFECT OF ALIGNMENT

The difference between aligned and random selection arises from the *inter-layer coherence* between successive subspaces, defined as

$$\alpha_b := \big\| P_{S_{b-1}^{(\mathrm{out})}} P_{S_b^{(\mathrm{in})}} \big\|_2 \in [0,1].$$

This coherence quantifies how well the selected subspaces of consecutive blocks overlap.

**Aligned selection.**   When the same subspace is consistently chosen across layers, we have $\alpha_b \approx 1$ for all $b$, i.e., the active coordinates of consecutive blocks are well aligned.

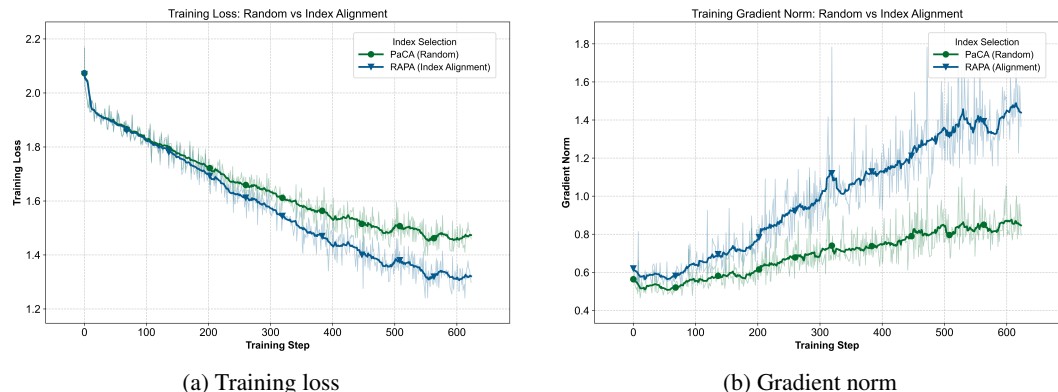

Figure 11: Training dynamics under PaCA (random) vs. RAPA(index alignment).

**Random selection.** When each block selects its subspace independently at random, the expected overlap is much smaller:

$$\mathbb{E}[\alpha_b] \approx \sqrt{k/d} < 1,$$

where $k$ is the active dimension and $d$ is the ambient dimension.

Using

$$\left\| P_{S_{b-1}^{(\text{out})}} J_b P_{S_b^{(\text{in})}} \right\|_2 \leq \|J_b\|_2 \, \alpha_b,$$

we can express the variance bound as

$$\text{Var}(g_0) \lesssim \Big( \prod_{b=1}^m \big( \|J_b\|_2^2 \alpha_b^2 \big) \Big) \text{Var}(g_m) + \sum_{i=1}^m \Big( \prod_{b=1}^{i-1} \big( \|J_b\|_2^2 \alpha_b^2 \big) \Big) \mathbb{E} \big\| P_{S_{i-1}^{(\text{out})}} \xi_i \big\|_2^2.$$

## J.4 INTERPRETATION

Intuitively, each coherence factor $\alpha_b$ scales both the variance propagated from the upper layers and the transmitted variance of all preceding noise terms. For *random* selection ($\alpha_b < 1$), the cumulative product $\prod_b \alpha_b^2$ becomes small, so all propagated components—including both signal and variance—are strongly attenuated. This leads to vanishing gradients and poor optimization signal-to-noise ratio (SNR) along deep pathways.

In contrast, for *aligned* selection ($\alpha_b \simeq 1$), the effective Jacobian norm $\rho_b$ remains close to $\|J_b\|_2$ while each $\mathbb{E}\|P_{S_{b-1}^{(\text{out})}} \xi_b\|_2^2$ is reduced approximately by the active dimension ratio $(k/d)$. Thus, alignment preserves the useful gradient signal while still controlling the total injected variance.

Although the numerical upper bound in the aligned case may appear larger (because $\alpha_b$ is larger), it is in fact *tighter in practice*: it accurately reflects the controlled variance growth within the consistently selected subspace. Empirically, this aligned regime corresponds to higher gradient SNR and faster convergence during optimization.

As shown in Figure 11(a), index alignment reduces the training loss faster than random selection, while Figure 11(b) shows that it also maintains a larger gradient norm in deeper training steps.

## K LIMITATION

While RAPA demonstrates strong empirical accuracy and training efficiency across a variety of fine-tuning tasks, several limitations remain. First, our method is evaluated solely in the context of fine-tuning pretrained language models; its applicability to other adaptation paradigms, such as continual learning or multi-stage pretraining, remains an open question. Second, although RAPA improves gradient propagation via structured index alignment, its effectiveness may be sensitive to model architecture and depth, particularly in transformer variants that diverge from standard residual block structures. Finally, RAPA employs a fixed global index set shared across layers. While this promotes implementation simplicity and structural coherence, it may limit the potential for layer-specific specialization. Exploring more flexible alignment strategies that adaptively tailor index sets per layer is a promising direction for future work.

## L BROADER IMPACTS

### L.1 MULTI-RAPA: EFFICIENT MULTI-ADAPTER SERVING WITHOUT MERGING

Recent works such as S-LoRA (Sheng et al., 2023) has highlighted the importance of supporting multiple fine-tuned adapters simultaneously in deployment environments. These systems aim to improve inference scalability by avoiding the need to merge adapter weights back into the base model, dynamically switching between LoRA modules to support various tasks.

Our proposed method, RAPA, complements this trend by further reducing the memory and computational demands of each adapter. Unlike LoRA-based methods that apply low-rank updates across all input dimensions, RAPA fine-tunes only a small square-aligned subset of weights. This makes each adapter lighter in both parameter count and run-time footprint, which in turn enables more adapters to be hosted concurrently in constrained environments. During inference, RAPA requires fewer compute and memory resources per step, an advantage particularly relevant to multi-tenant inference servers, on-device deployment, and cost-sensitive applications.

In summary, RAPA not only improves parameter-efficient fine-tuning but also enables more scalable, sustainable, and inclusive model serving, advancing the field of adaptive language modeling.

### L.2 MOE-COMPATIBLE ADAPTATION WITH RAPA

RAPA supports Mixture-of-Experts (MoE) architectures by allowing each expert to update only a small square-aligned subspace of weights, requiring only 0.1 to 0.5% additional parameters per expert. This results in significantly reduced memory and computation overhead during both training and inference. When combined with sparse activation of MoE, the active parameter count per forward pass drops further, enabling low-latency and memory-efficient adaptation. Previous work such as MixLoRA (Li et al., 2024) and Mixtral (Jiang et al., 2024a) shows that lightweight adapters can improve multitask accuracy without compromising efficiency; RAPA achieves similar gains with even smaller experts and no merging overhead. Since each expert is only a few megabytes, hundreds of domain-specific RAPA experts can be held in memory simultaneously, supporting scalable and sustainable deployment. This design reduces entry barriers for low-resource settings, facilitates localization, and mitigates interference between tasks, offering a modular and future-proof approach to continuous customization of large models.

