# OpenReview forum: "RAPA: Recursively Aligned Pathway Adaptation of Large Language Models"
_ICLR.cc/2026/Conference — Submitted to ICLR 2026_

### Official Review · Reviewer_uCmc · 2025-10-27

**Soundness:** 2
**Presentation:** 3
**Contribution:** 2
**Rating:** 4
**Confidence:** 4

**Summary:**

This paper proposes RAPA, which is an adapter-free PEFT method for LLM fine-tuning. The core idea is to (i) reallocate fixed parameter budget into a balanced square submatrix per linear layer rather than LoRA’s low-rank adapters (such that total trainable params unchanged) and (ii) align the selected row/column index sets recursively across layers and through residual connections, so that gradients propagate along trainable pathway. The trainable square submatrix yields higher effective rank (compared to LoRA), and avoids adapter induced activation overheads. Empirically, RAPA matches or outperforms strong adapter-based (LoRA/DoRA) and adapter-free (PaCA) baselines.

**Strengths:**

1, Cross-layer index-aligned subspace tuning is a novel idea, backed up by explanations using backprop for why aligned indices improve gradient consistency.
2, Balanced submatrix offers higher ranks under equal trainable params budget
3, The paper provides substantial evaluation which show RAPA outperforms baselines under equal parameter budget

**Weaknesses:**

1, While the squared matrix maximizes the trainable matrix’s rank in theory, in practice it’s not clear why “rank” should be the only direction to optimize for. Explanations for why this optimality aligns with practical goals are absent.
2, The paper focuses on dense arch. MoE, another popular choice of LLM arch, is underexplored. Especially, it’s non-trivial and unclear whether router and expert FFN admit similar training dynamics and the same technique is applicable.
3, Different modules (such as kqv projections VS FFN) could favor different shaped submatrix or alignment mechanisms, but this is not well understood with the existing analysis. It’s also not clear how the embedding matrix (and final projection) is handled; considering its completely different function and large proportion of memory it takes, handling embedding is a subtle and non-trivial topic which was not touched.
4, The paper covers adapter based and adapter free methods, but left out one important area based on side-tuning, which relies on additional trainable parameters without triggering non-linear activations through backbone by design. Please include references to this line of work:
	Read: Recurrent adaptation of large transformers, John Nguyen, Sid Wang, Ke Li, and Carole-Jean Wu. 2023.
        Side-tuning: a baseline for network adaptation via additive side networks. Jeffrey O Zhang, Alexander Sax, Amir Zamir, Leonidas Guibas,and Jitendra Malik 2020

**Questions:**

1, While square trainable matrix maximizes rank under a fixed budget, could the authors explain why maximizing rank alone is the right optimization objective in practice? Are there either empirical or theoretical insights connecting rank-optimality to downstream task performance, beyond heuristic intuition presented in the text?
2, How does RAPA extend to MoE architectures, where routers and expert FFNs may follow different learning dynamics? Do the authors expect the same alignment mechanism to apply to both router and expert parameters, or are modifications necessary to maintain stability?
3, Could different modules (e.g., kqv projections vs. FFN layers) benefit from different submatrix shapes or alignment strategies? Additionally, how are token embeddings and the LM head treated in RAPA? Given their distinct functional roles and large parameter footprint, have the authors explored whether they require different handling?
4, The related-work discussion omits side-tuning based adapter-free methods that add parallel trainable subnetworks without modifying backbone nonlinearities (e.g., Zhang et al., 2020; Nguyen et al., 2023). Could the authors compare RAPA against or at least position it relative to this class of methods?
5, Have you tested on 34B/70B (or larger) models? Given your kernel-launch argument, do you anticipate the relative efficiency gap to widen or narrow at scale?
6, Given the recursive nature of your approach, can you refine your theoretical analysis by deriving bound on gradient variance iteratively along the pathway?

---

> ### Author Response · Authors · 2025-11-20
> **Q4.1) Why “rank” should be the only direction to optimize for**
>
> Thank you for raising this point. We agree that rank should not be viewed as the *only* objective in designing a PEFT update, and we will clarify this in the revised version. Our intention in Section 3.2.1 was more modest: among all submatrices with the same parameter budget, a balanced square block both
>
> - maximizes the rank of the trainable subspace, and
> - maximizes the opportunity for cross-layer index reuse, which RAPA then exploits via pathway alignment.
>
> **(1) Rank and shape as a *guiding* criterion, supported by ablations**
> Rather than relying on theory alone, we also study this effect empirically. In Sec. 4.1 / Fig. 5 we sweep a wide range of row/column ratios at fixed parameter count. Under random index selection, MMLU accuracy systematically improves as the shape becomes more balanced:
>
> > **Table R4.1: Performance comparison as a function of selected weight shape and index alignment**
> >
> >
> >
> > | Shape (row/col %) | Random selection | + RAPA alignment |
> > | --- | --- | --- |
> > | 0 / 100 (column-only) | 45.6% | 47.3% |
> > | 50 / 50 (square) | **47.4%** | **48.9%** |
> > | 100 / 0 (row-only) | 43.2% | 45.1% |
>
> The 50/50 configuration (highest rank for a given budget) is consistently best:
>
> - vs. column-only: +1.8 points (random) / +1.6 points (aligned),
> - vs. row-only: +4.2 / +3.8 points.
>
> These ablations indicate that, *for a fixed number of parameters*, moving toward a higher-rank, shape-balanced subspace is strongly correlated with better task performance—even before we consider cross-layer alignment.
>
> **(2) Rank is coupled with optimization-aware alignment, not used in isolation**
>
> In practice, rank serves as a **guiding proxy**, and our design always couples the square selection with RAPA’s structural index alignment, which directly targets optimization dynamics.
> Sec. 3.2.2 and Appendix B show that aligned indices create coherent gradient pathways across residual blocks, yielding shortcut terms in the backpropagated gradient (Eqs. (6)–(7), (17), (20)). Empirically, Fig. 4 and Table 1 demonstrate that, under the *same* parameter budget:
>
> - the aligned variant converges faster in training loss, and
> - MMLU improves from 51.7% (random selection) to 52.3% (Block+Global alignment).
>
> Appendix E (moved from C) further confirms that this improvement in convergence is consistent across random seeds.

---

> > ### Author Response · Authors · 2025-11-20
> > **Q4.2) How does RAPA extend to MoE architectures? (1/2)**
> >
> > We appreciate the reviewer’s observation that our main experiments focus on dense transformers. We agree that MoE is an important architecture, and we will clarify both (i) how our index-alignment mechanism extends to MoE, and (ii) preliminary empirical evidence on MoE models.
> >
> > Motivated by recent MoE–PEFT frameworks such as LoRA-MoE and Nexus, which freeze the router and only adapt a small subset of expert or backbone weights, and consistent with recent MoE overviews (Zhang et al., 2025), we apply RAPA to the attention layers and expert FFNs while keeping the router parameters frozen as in the baseline MoE model (i.e., no additional alignment or modification).
> >
> > > Zhang, Danyang, et al. "Mixture of experts in large language models." *arXiv preprint arXiv:2507.11181* (2025).
> > >
> >
> > ### (1) MoE residual blocks still admit the same pathway structure
> >
> > Consider a standard MoE FFN block at layer $b$:
> >
> > $$
> > x^{(b+1)} = x^{(b)} + \sum_{e=1}^{E} p_e^{(b)}(x^{(b)})\, F_e^{(b)}(W^{(b,e)}, x^{(b)}),
> > $$
> >
> > where $p_e^{(b)}(x^{(b)})$ are router probabilities and $F_e^{(b)}$ is the expert FFN.
> >
> > In the MoE version of RAPA, we choose a common index set $J \subset \{1,\ldots,d\},|J| = k,$  and restrict **all experts in this block** to update only the coordinates in $J$. That is, each expert FFN operates on $x_J^{(b)}$ and uses weights $W_{J,J}^{(b,e)}$. Focusing on the $J$-subspace, the block becomes:
> >
> > $$
> > x_J^{(b+1)} = x_J^{(b)} + \sum_{e=1}^{E} p_e^{(b)}(x^{(b)})\, F_e^{(b)}(W^{(b,e)}_{J,J}, x_J^{(b)})
> > =: x_J^{(b)} + \tilde{F}^{(b)}(\tilde{W}^{(b)}, x_J^{(b)}).
> > $$
> >
> > If we stack $m$ such MoE blocks and apply the chain rule, we obtain (details will be added in the appendix):
> >
> > $$
> > \frac{\partial \mathcal{L}}{\partial x_J^{(b)}}= \frac{\partial \mathcal{L}}{\partial x_J^{(b+m)}}
> > \Bigg[ I + \frac{\partial}{\partial x_J^{(b+m)}} \sum_{t=0}^{m-1} \tilde{F}^{(b+t)}(\tilde{W}^{(b+t)}, x_J^{(b+t)}) \Bigg].
> > $$
> >
> > Crucially, as in the dense case, there is an **identity term on $x_J^{(b)}$** plus additional terms from the experts. Thus, even with MoE, aligning the same index set $J$ across MoE blocks preserves the **direct residual shortcut** in the $J$-subspace and the same “pathway” structure that our dense analysis relies on.
> >
> > ### (2) Layer-wise index alignment inside MoE experts
> >
> > For an expert $e$, we write the FFN as
> >
> > $$
> > f_e(x) = W_2^{(b,e)} \,\phi\big(W_1^{(b,e)} x\big).
> > $$
> >
> > To apply layer-wise alignment, we choose indices $(I,J)$ such that
> >
> > - $W_1^{(b,e)}$ is trainable only on the block $(I,J)$, and
> > - $W_2^{(b,e)}$ is trainable only on the block $(J,I)$,
> >
> > with **the same $I,J$ shared across all experts** in that layer. Restricting to $x_J$, the MoE FFN contribution to the residual becomes:
> >
> > $$
> > x_J^{(b+1)}= x_J^{(b)}+ \sum_{e} p_e^{(b)}(x^{(b)}) \,W_{2,J,I}^{(b,e)} \,\phi\big(W_{1,I,J}^{(b,e)} x_J^{(b)}\big).
> > $$
> >
> > For an expert’s first linear layer, the gradient takes the form
> >
> > $$
> > \frac{\partial \mathcal{L}}{\partial W_{1,I,J}^{(b,e)}} = \frac{\partial \mathcal{L}}{\partial x_J^{(b+1)}} \cdot p_e^{(b)}(x^{(b)}) \cdot \frac{\partial f_e}{\partial W_{1,I,J}^{(b,e)}}.
> > $$
> >
> > When we expand $\partial f_e / \partial W_{1,I,J}^{(b,e)}$, we obtain factors of the form
> >
> > $$
> > W_{2,J,I}^{(b,e)} \, x_J^{(b)},
> > $$
> >
> > which is **exactly analogous** to the dense case, up to the multiplicative router weight $p_e^{(b)}(x^{(b)})$. Thus, layer-wise index alignment within experts preserves the same linear “pathway” coupling between the input $x_J^{(b)}$ and the output residual $x_J^{(b+1)}$; the MoE router just reweights each expert’s contribution.
> >
> > ### (3) Router gradients and stability
> >
> > If we include the router gradient, the derivative of $\tilde{F}^{(b)}$ with respect to $x_J^{(b)}$ contains both
> >
> > - terms where $p_e^{(b)}$ acts as a “scale factor” on expert gradients, and
> > - additional terms of the form
> >
> >     $$
> >     \sum_e \frac{\partial p_e^{(b)}(x^{(b)})}{\partial x_J^{(b)}} F_e^{(b)}(W_{J,J}^{(b,e)}, x_J^{(b)}),
> >     $$
> >
> >
> > which are **still defined inside the same $J$-subspace**.
> >
> > These router-related terms add extra contributions but do not remove the identity shortcut on
> >
> > $x_J^{(b)}$, and they do not require a different alignment rule. In our experiments, we keep the router parameters’ training scheme identical to the MoE baseline (no special RAPA constraint on router weights), and we did not observe any instability attributable to RAPA.
> >
> > Hence, the core residual alignment argument (identity + aligned MoE updates in $J$) continues to hold, while the router acts as a smooth, subspace-preserving reweighting mechanism.

---

> > > ### Author Response · Authors · 2025-11-20
> > > **Q4.2) How does RAPA extend to MoE architectures? (2/2)**
> > >
> > > ### (4) MoE experiment result
> > >
> > > To check that this analysis is not purely theoretical, we also conducted preliminary experiments on a MoE LLM (DeepSeek-MoE-16B). Under the same parameter budget, aligning the expert FFN indices across layers (and across experts) leads to:
> > >
> > > - lower validation perplexity compared to random expert indices, and
> > > - a small but consistent improvement in downstream metrics (MMLU accuracy).
> > >
> > > **Table R4.2: MMLU accuracy on the MoE architecture under different index alignment strategies**
> > >
> > > | **Method** | **PPL (↓)** | **Human.** | **STEM** | **Social.** | **Other** | **Average** |
> > > | --- | --- | --- | --- | --- | --- | --- |
> > > | Baseline | 8.71 | 41.3 | 36.9 | 51.2 | 51.7 | 44.8 |
> > > | PaCA (Square, Random) | 4.34 | 47.8 | 40.8 | 57.9 | 57.3 | 50.5 |
> > > | RAPA(Square, Aligned) | 3.74 | 48.9 | 42.1 | 59.1 | 59.5 | 51.9 |
> > >
> > > We included these MoE results, together with the full derivation sketched above, in the appendix of the revised version. While a full MoE benchmark suite is beyond the scope of this submission, these results suggest that the **pathway-alignment mechanism extends naturally from dense FFNs to MoE experts**, with the router acting as a smooth, subspace-preserving reweighting.

---

> > > > ### Author Response · Authors · 2025-11-20
> > > > **Q4.3) Module-specific shapes and embeddings / LM head**
> > > >
> > > > We thank the reviewer for pointing out that different modules (Q/K/V projections vs. FFN) may prefer different submatrix shapes, and for raising the question of how embeddings and the final projection are handled.
> > > >
> > > > ### (1) Module-specific shapes: attention vs. FFN
> > > >
> > > > We agree that attention and FFN layers have different native structures. In LLaMA2-7B, for example, the FFN block is wide-then-narrow (e.g., `gate_proj: 4096 → 11008`, `down_proj: 11008 → 4096`, with $d_{\text{ff}} \approx 2.7 \, d_{\text{model}}$), whereas Q/K/V/O projections are nearly square in the model dimension.
> > > >
> > > > RAPA does **not** enforce a single universal shape across all modules. Instead:
> > > >
> > > > - For **attention projections** (Q/K/V/O), we use **balanced shapes** in the shared hidden dimension, matching the square-like structure of these matrices.
> > > > - For **FFN layers** (up/gate/down), we keep the **same input index set** aligned with the residual stream but allocate a proportionally larger output index set, scaled by $d_{\text{ff}} / d_{\text{model}}$. In other words, we respect the natural expansion/contraction structure of the FFN while preserving cross-layer index alignment on the residual dimension.
> > > >
> > > > We explicitly compared a naïve “all square” variant to this architecture-aware RAPA design under the **same parameter budget** on LLaMA2-7B:
> > > >
> > > > **Table R4.3: Effect of FFN weight shape on performance under the same number of trainable weights**
> > > >
> > > > | Shape strategy | # Trainable params | 5-shot MMLU |
> > > > | --- | --- | --- |
> > > > | RAPA (all-square) | 20M | 52.4% |
> > > > | RAPA (module-aware) | 20M | **53.0%** |
> > > >
> > > > Thus, even within a fixed subspace rank/budget, **respecting the native shape of FFN vs. attention** provides a measurable gain. This is also consistent with prior work showing that FFN layers store much of the model’s factual associations (e.g., Meng et al., 2022), which motivates allocating a slightly richer subspace to FFN expansions while keeping the residual-aligned indices consistent.
> > > >
> > > > > Meng, Kevin, et al. "Locating and editing factual associations in gpt." *Advances in neural information processing systems* 35 (2022): 17359-17372.
> > > > >
> > > >
> > > > ### (2) Embeddings and LM head
> > > >
> > > > We agree that token embeddings and the tied LM head differ from other modules in both their functional roles and large parameter footprint, so it is natural to ask whether they should be treated differently.
> > > >
> > > > In our experiments, we follow the **standard PEFT setup** and **keep both the input embedding matrix and the LM head frozen** for all methods, including RAPA and baselines, for two reasons:
> > > >
> > > > 1. **Fair comparison and efficiency.** Most PEFT algorithms we compare against (LoRA, IA3, DoRA, AdaLoRA, MosLoRA, Prefix-tuning, BitFit, etc.) are typically evaluated with frozen embeddings and output head. Turning on embedding / LM-head updates would change the memory and compute profile and break the “same setting” comparison we aim for.
> > > > 2. **Stability with weight tying.** In LLaMA-style models the LM head is usually tied to the token embeddings. Aggressively fine-tuning this large matrix can degrade training stability and obscure the effect of the *structured subspace* we are studying (RAPA), especially under low-budget PEFT regimes.
> > > >
> > > > The exact set of trainable modules for each method (linear projections, biases, embeddings) is already listed in our hyperparameter tables in the appendix; these settings are shared across all baselines and RAPA, ensuring that we operate in the same parameter regime and that the treatment of embeddings / LM head is deliberate rather than an omission.

---

> > > > > ### Author Response · Authors · 2025-11-20
> > > > > **Q4.4) Position of Side-tuning / READ compared to RAPA**
> > > > >
> > > > > We thank the reviewer for pointing us to the side-tuning line of work and papers.
> > > > >
> > > > > Side-tuning (Zhang et al., 2020) and READ (Nguyen et al., 2023) follow a design where a **separate, lightweight side network** is trained in parallel to a frozen backbone, and the final prediction is obtained by an additive or gated combination of the backbone and side-network outputs.
> > > > >
> > > > > **Table R4.4: Comparison of different finetuning methods**
> > > > >
> > > > > | **Comparison Metric** | **Adapter-based PEFT (LoRA)** | **Side-Tuning / READ (Side-Network)** | **Adapter-free PEFT (RAPA)** |
> > > > > | --- | --- | --- | --- |
> > > > > | **Core Structure** | Attaches **low-rank linear modules** in **parallel** to the frozen backbone weights. | Operates a **separate, lightweight network** in **parallel** to the backbone. | Updates a **sparse subset** of the base weights following **aligned pathways**. |
> > > > > | **Forward Overhead (Training)** | **High** (~40% additional Latency; **3x** more GEMM kernel calls) | **High** (Structural overhead from executing a separate parallel network) | **Very Low** (**~4%** additional Latency; kernel call count remains nearly the same as baseline) |
> > > > > | **Forward Overhead (Inference)** | **Zero** (Achieved via **Merge**) | **High** (Non-mergeable, must run the Side Network *always*) | **Zero** (Achieved via **Merge** or inherent Adapter-free design) |
> > > > > | **Backward Overhead (Training)** | **Low** (Efficient gradient calculation for small $\Delta W$; requires storing full activation) | **Low** (Gradients only for side network parameters) | **Low** (Efficient gradient calculation for sparse $W_{rapa}$) |
> > > > > | **Mergeability (Inference Optimization)** | **Possible (High)** (Linear operation) | **Impossible/Difficult (Low)** (Structural mismatch) | **Possible (High)** (Sparse $\Delta W$ is overwritten onto $W_{base}$) |
> > > > > | **Structural Category** | **Additive (Adapter-based)** | **Additive (Adapter-based)** | **Adapter-free** (Avoids LoRA's structural overhead) |
> > > > >
> > > > > **In terms of computational structure, we view these methods as sharing the same "additive" nature as LoRA and other parallel adapters.** While the specific architecture of the side branch differs (e.g., a small neural network in Side-Tuning vs. low-rank matrices in LoRA), both approaches introduce auxiliary parameters and perform parallel computations $(h = W_{fixed}x + f_{side}(x))$ alongside the frozen backbone.
> > > > > However, a practical limitation of Side-Tuning–style approaches is that the **side network cannot be cleanly merged back into the pretrained backbone after training**; the model must continue to evaluate the side branch at inference time. Consequently, these methods cannot exploit the key advantage of many adapter-based or adapter-free approaches—namely, that the adapted parameters can be merged into the backbone so that inference incurs (almost) no additional latency or memory
> > > > >
> > > > > Our focus in this paper is somewhat complementary:
> > > > >
> > > > > - On the **adapter-based side**, we compare against methods such as **LoRA, DoRA, MosLoRA, PiSSA, and VeRA**, which also add small trainable modules around a frozen backbone. In these methods, the extra parameters are implemented as low-rank (or low-dimensional) linear adapters attached to existing projections, rather than as a full side network.
> > > > > - On the **adapter-free side**, RAPA and PaCA aim to select and update a sparse subset of the *existing* backbone weights without introducing new forward paths. Our main contributions (square-shaped subspaces and recursive index alignment) are about **where inside the backbone** to place the parameter budget, rather than how to design an external side network.
> > > > >
> > > > > From the perspective of our paper’s main focus—**parameter efficiency and inference overhead**—Side-Tuning/READ and LoRA-based methods fall into the same category of "additive approaches" that incur extra memory and computational cost for the side branches.
> > > > > In contrast, our proposed method, RAPA, is **structurally different from both**. RAPA aims to select and update a sparse subspace *within* the existing backbone weights. This design avoids introducing any external side networks or parallel branches, thereby maintaining the original inference architecture without the latency overhead associated with additive modules.
> > > > >
> > > > > We explicitly acknowledge Side-Tuning and READ in our related work section, classifying them alongside LoRA as parallel additive adaptation methods. **In contrast to these additive approaches,** RAPA operates directly on the backbone without introducing any parallel branches.
> > > > >
> > > > > - Zhang et al., 2020. *Side-Tuning: A Baseline for Network Adaptation via Additive Side Networks.*
> > > > > - Nguyen et al., 2023. *READ: Recurrent Adaptation of Large Transformers.*

---

> ### Author Response · Authors · 2025-11-20
> **Q4.5) Scaling of Overhead/Performance to Larger Models**
>
> ### (1) Scaling of Overhead to Larger Models
>
> To examine how the overhead pattern in Section 3.1 behaves across different widths, we repeated our profiling on two decoder-only Transformers with hidden dimensions $h_{\text{dim}} = 1024$ (smaller) and $h_{\text{dim}} = 8192$ (wider). For each model, we compare a frozen baseline, an adapter-based (AB) method, and an adapter-free (AF) counterpart under exactly the same benchmarking protocol as in the main-text overhead analysis.
>
> As summarized in the new appendix I (Fig. 11), the qualitative behavior is consistent across widths. In both regimes, the AB method triggers about $3 \times$ more GEMM kernels than the baseline, translating into roughly 35–45% forward-time overhead. By contrast, the AF method closely tracks the baseline in both kernel count and runtime, incurring only a few percent overhead.
>
> We further sweep batch size, sequence length, and adapter rank $r$, and report the kernel-count and GEMM-time *ratios* (AB / baseline). The overhead is largely insensitive to $h_{\text{dim}}$ and $r$, and instead grows mainly with batch size and sequence length due to the structural cost of launching additional small kernels. Since the number and ordering of adapter kernels scale primarily with depth rather than width, there is no mechanism at larger scale that would narrow this structural gap. We therefore expect the relative efficiency gap between AB and AF to remain at least as large—if not slightly larger due to increased depth—when moving to 34B/70B models, while AF remains near parity with the frozen baseline.
>
> ### (2) Performance Evaluation on Larger-Scale Models
>
> As suggested by the reviewer, we additionally tested our method on a larger model. Specifically, we fine-tuned **LLaMA-2-70B** on the **Metamathqa-40K** dataset under the same adapter-free setting. This experiment confirms that RAPA is applicable and effective at the 70B scale as well: even on this larger backbone, aligned square index selection yields consistent gains over the baseline and the PaCA variant under the same parameter budget (see Table R4.5). These results indicate that the benefits of our alignment mechanism are not limited to small models, but extend to substantially larger architectures.
>
> **Table R4.5: Result of GSM8K 0-Shot Evaluation (Metamathqa-40K Fine-tuning)**
>
> | Method | Training Data | Evaluation Metric | GSM8K (0-shot) Accuracy (%) |
> | --- | --- | --- | --- |
> | **Baseline** (Frozen) | N/A | Exact Match | 16.2 |
> | **PaCA (Square, Random)** | Metamathqa-40K | Exact Match | 53.3 |
> | **RAPA (Square, Aligned)** | Metamathqa-40K | Exact Match | **54.8** |

---

> ### Author Response · Authors · 2025-11-20
> **Q4.6) Recursive Gradient-Variance Bound Along the Aligned Pathway**
>
> We thank the reviewer for this suggestion. In response, we derive a **recursive variance bound** along the aligned pathway and added the full derivation as Appendix J.
>
> We model the backpropagated gradient through block $b$ as
>
> $$
> g_{b-1} = P_{S_{b-1}^{(\mathrm{out})}} \big(J_b P_{S_b^{(\mathrm{in})}} g_b + \xi_b\big),
> $$
>
> where $J_b$ is the local Jacobian, $P_{S_b^{(\mathrm{in})}}, P_{S_{b-1}^{(\mathrm{out})}}$ are the input/output projections onto the $k$-dimensional selected subspaces, and $\xi_b$ is a zero-mean noise term (mini-batch, normalization, quantization, etc.). Taking conditional variance and unrolling over mmm blocks yields a product–sum bound of the form
>
> $$
> \mathrm{Var}(g_0) \lesssim \Big(\prod_{b=1}^{m}\rho_b^2 \Big)\mathrm{Var}(g_m)+\sum_{i=1}^{m}\Big(\prod_{b=1}^{i-1}\rho_b^2 \Big)\mathbb{E}\big\|P_{S_{i-1}^{(\mathrm{out})}}\xi_i\big\|_2^2,
> $$
>
> where $\rho_b = \|P_{S_{b-1}^{(\mathrm{out})}} J_b P_{S_b^{(\mathrm{in})}}\|_2$ is the effective spectral norm at block $b$.
>
> The key quantity that distinguishes **aligned** vs. **random** selection is the inter-layer coherence
>
> $$
> \alpha_b := \big\|P_{S_{b-1}^{(\mathrm{out})}} P_{S_b^{(\mathrm{in})}}\big\|_2 \in [0,1],
> $$
>
> which measures how much consecutive subspaces overlap. For aligned selection, we have $\alpha_b \approx 1$, whereas under random selection a simple model gives $\mathbb{E}[\alpha_b] \approx \sqrt{k/d} < 1$. Using $\|P_{S_{b-1}^{(\mathrm{out})}} J_b P_{S_b^{(\mathrm{in})}}\|_2 \le \|J_b\|_2 \alpha_b$, the bound becomes
>
> $$
> \mathrm{Var}(g_0) \lesssim \Big(\prod_{b=1}^{m}(\Vert J_b\Vert_2^2\alpha_b^2)\Big)\mathrm{Var}(g_m) + \sum_{i=1}^{m} \Big(\prod_{b=1}^{i-1}(\Vert J_b\Vert_2^2\alpha_b^2)\Big)\, \mathbb{E}\big\Vert P_{S_{i-1}^{(\mathrm{out})}}\xi_i\big\Vert_2^2.
> $$
>
> Intuitively:
>
> - For **random** selection, $\alpha_b < 1$ on average, so the product $\prod_b \alpha_b^2$ becomes small and attenuates **both** signal and variance, leading to vanishing gradients and a poor optimization SNR.
> - For **aligned** selection, $\alpha_b \simeq 1$, so the useful gradient signal is preserved along the pathway while the injected noise is still reduced roughly in proportion to the active dimension ratio $k/d$.
>
> This recursive bound formalizes the intuition that RAPA’s aligned subspaces maintain a more stable, higher-SNR gradient pathway than random selection, which is consistent with our empirical observation that aligned variants converge faster and achieve better downstream performance.

---

> > ### Comment · Reviewer_uCmc · 2025-11-27
> > **Response to authors**
> >
> > I thank the authors for the detailed rebuttal. The additional empirical evidence substantially clarifies several of my main concerns, particularly the shape-vs-rank ablations, the module-aware FFN experiments, and the preliminary MoE results. The recursive gradient-variance derivation is also a valuable addition that deepens the theoretical grounding of the method. While some conceptual questions remain open, especially concerning whether rank maximization alone is the right guiding principle across architectures, the overall submission has improved enough that I am raising my score. One practical note: the LaTeX in the final block of the rebuttal (the recursive variance bound) does not compile in its current form; I encourage the authors to ensure the final camera-ready version includes the fully working LaTeX so readers can reproduce the derivations.

---

> > > ### Author Response · Authors · 2025-11-28
> > > **Appreciation for your follow-up review and suggestions**
> > >
> > > Thank you very much for your thoughtful follow-up and for raising your score based on the revised rebuttal and manuscript. We truly appreciate the time you invested in re-reading our response and in commenting on the new experiments and theory.
> > >
> > > Following your suggestion regarding the **recursive gradient-variance bound**, we have now included **a concise summary of this result in the main text (page 6) and we provide the complete derivation in Appendix J**, carefully checking that all expressions compile correctly and can be reproduced by readers.
> > >
> > > We are also grateful that you highlighted the value of the shape-vs-rank ablations, the module-aware FFN experiments, and the preliminary MoE results—your comments pushed us to organize and present these components more clearly, which we believe makes the paper more understandable and useful to future readers.
> > >
> > > Thank you again for your constructive engagement throughout the review and discussion process.
> > >
> > > Best regards,
> > >
> > > The Authors

---

### Official Review · Reviewer_Ptgb · 2025-10-31

**Soundness:** 3
**Presentation:** 2
**Contribution:** 3
**Rating:** 6
**Confidence:** 4

**Summary:**

This paper proposes RAPA (Recursively Aligned Pathway Adaptation), a new adapter-free parameter-efficient fine-tuning (PEFT) algorithm for large language models (LLMs). It selects balanced square submatrices to maximize expressive capacity under a fixed parameter budget and recursively aligns trainable indices across layers, blocks, and residual connections. This alignment aims to create coherent gradient pathways, reducing discontinuities and accelerating convergence. Experiments in the paper show RAPA matching or surpassing baselines like LoRA, DoRA, and PaCA on benchmarks such as MMLU, with lower memory and training time.

**Strengths:**

- Efficiency Gains: RAPA minimizes training overhead by avoiding adapter modules, which can lead to fewer kernel launches and lower activation memory. Experiments indicate it uses around 41-53 GB of memory for models like LLaMA2-7B/13B, often outperforming adapter-based methods in speed and resource use.
- Actionable Ablation Studies: Figure 5 on p.7 demonstrates that shaping the row/column ratio for the tuned submatrix and applying index alignment both improve task performance, supporting the design decisions empirically as well as theoretically.
- Performance Improvements: By aligning pathways recursively, RAPA stabilizes gradient propagation, leading to faster convergence and higher accuracy on benchmarks such as MMLU (e.g., 54.6% average for LLaMA2-7B). This structured selection of balanced submatrices enhances representational capacity under fixed parameter budgets.

**Weaknesses:**

- Reliance on Random Index Selection: While RAPA's primary contribution is the recursive alignment of pathways, the initial selection of this pathway is based on a randomly chosen index set $I$ (Appendix E). A concern arising from this design is that a purely random selection, even if structurally coherent, may not be optimal. It seems likely that this method could miss weights that are critical for task adaptation, potentially limiting its performance relative to importance-driven or calibrated approaches.
- Questionable Statistical Significance of Gains: The stability of the method's reported gains warrants closer inspection. Although Appendix A analyzes robustness to random seeds, the reported fine-tuning performance across four seeds shows a non-trivial performance spread (53.7% to 54.3%, +0.6%, from Table 5). This fluctuation resulting from the random seed choice is comparable in magnitude to the performance improvement claimed over prior work, such as PaCA (+0.6% on LLaMA2-13B, Table 2). This raises questions about the statistical significance of the reported gains and suggests that the method's advantages might be highly sensitive to the specific random seed chosen for index selection.
- Unsubstantiated Contribution of “Rank Expansion”: This benefit is not experimentally disentangled from the paper's other core contribution: index alignment. It is plausible that the observed performance gains are predominantly attributable to the index alignment strategy, rather than the putative “rank expansion”. The paper would be significantly strengthened by an ablation study that isolates these two effects (e.g., by applying RAPA's “Rank Expansion” parameterization to other adapter-free methods like PaCA.)

**Questions:**

- Can the method be extended or modified to support dynamic, input- or data-dependent index selection, without incurring the calibration cost that motivates RAPA’s simplicity? What are the trade-offs?
- In Figure 3, the current diagram does not clearly visualize how these three different types of alignment are implemented or how they relate to each other. The labels (1), (2), and (3) are present, but their graphical representation is ambiguous. Could the authors revise it to provide a more explicit and differentiated illustration of each of the three alignment strategies? This would significantly improve the paper's clarity and help readers grasp the precise mechanics of the RAPA method.
- Could the authors comment on the method's sensitivity to other hyperparameters and design choices that were not tested? For example, potential confounders such as the distribution of selected indices (e.g., contiguous vs. sparse) are not explored. This leaves a gap in understanding the method's robustness to its own hyperparameter changes.

---

> ### Author Response · Authors · 2025-11-20
> **Q3.1) Reliance on Random Index Selection (1/2)**
>
> We sincerely appreciate your careful reading of our paper and your insightful feedback.
> Below, we provide detailed responses addressing the strengths, limitations, and questions you pointed out.
>
> We agree that a **purely random choice of the global index set is unlikely to be provably optimal**, and it is natural to ask whether importance-aware or calibrated selection could do better. We therefore explored two directions beyond naive randomness and report why we ultimately adopt simple random global indices in this work.
>
> ### (1) Sensitivity probe vs. final performance (Appendix A)
>
> Empirically, we find that **pre-finetuning “sensitivity” of a weight subset can vary quite a lot across random index sets (std ≈ 3.5 points in MMLU accuracy), but after finetuning with RAPA, the variation in final accuracy collapses to only ≈ 0.2 points**. In other words, although some index sets look much more “important” than others under a sensitivity probe, **RAPA’s training compensates for these differences**, and the final downstream performance is essentially insensitive to which random global index set is chosen.
>
> To probe this, we first measured the **pre-finetuning sensitivity** of a global index set to the downstream task:
>
> - For each random seed, we select a global index set as in RAPA.
> - **Before finetuning**, we **mask those weights to zero** and measure the drop in MMLU. A larger drop means the chosen indices are more “sensitive” for the downstream task.
>
> This yields Table R3.1:
>
> > Table R3.1: Weight-masking test (set to zero) for sensitivity probe (higher drop ⇒ more sensitive)
> >
> >
> > (numbers are absolute accuracy after masking; lower = larger degradation)
> >
> > | Setting | Hums. | STEM | Social. | Other | Avg. |
> > | --- | --- | --- | --- | --- | --- |
> > | Seed #1 | 34.0 | 31.8 | 38.5 | 42.8 | 36.4 |
> > | Seed #2 | 39.9 | 35.9 | 48.3 | 49.9 | 43.0 |
> > | Seed #3 | 39.5 | 38.3 | 47.4 | 49.1 | 43.1 |
> > | Seed #4 | 40.4 | 37.5 | 49.6 | 50.6 | 44.0 |
> > | **Mean ± SD** | **38.5 ± 3.0** | **35.9 ± 2.9** | **46.0 ± 5.0** | **48.1 ± 3.6** | **41.6 ± 3.5** |
>
> Here, **Seed #1** clearly corresponds to a more “damaging” index choice (larger average drop), while Seed #4 looks much less harmful. In other words, random global indices indeed differ in **how much pre-finetuning sensitivity** they capture.
>
> However, when we **actually fine-tune** RAPA using these same index sets, the picture changes:
>
> > Table R3.2: After RAPA fine-tuning with the same indices
> >
> >
> >
> > | Setting | Hums. | STEM | Social. | Other | Avg. |
> > | --- | --- | --- | --- | --- | --- |
> > | Seed #1 | 50.2 | 44.1 | 63.2 | 60.4 | 54.0 |
> > | Seed #2 | 51.3 | 43.2 | 62.6 | 60.9 | 54.1 |
> > | Seed #3 | 51.2 | 42.6 | 62.1 | 60.5 | 53.7 |
> > | Seed #4 | 51.2 | 44.3 | 62.8 | 60.6 | 54.3 |
> > | **Mean ± SD** | **51.0 ± 0.5** | **43.6 ± 0.8** | **62.7 ± 0.5** | **60.6 ± 0.2** | **54.0 ± 0.2** |
>
> Despite the **3.5-point standard deviation** in the sensitivity probe (Table R3.1 Avg.), the **final finetuned accuracy varies by only ~0.2 points** across seeds (Table R3.2 Avg.). Notably, Seed #1—initially the most “sensitive” choice—ends up essentially indistinguishable from the others after training.
>
> Taken together, these results suggest that:
>
> - Pre-finetuning sensitivity *does* vary across random index sets (so random selection is not “perfect”),
> - but **training quickly re-optimizes within the chosen aligned subspace**, and the initial sensitivity differences largely wash out.
>
> Empirically, under our parameter budgets, **we did not see a strong, consistent benefit from trying to preferentially choose “more sensitive” indices** as opposed to simply using random global indices plus cross-layer alignment.

---

> > ### Author Response · Authors · 2025-11-20
> > **Q3.1) Reliance on Random Index Selection (2/2)**
> >
> > ### (2) Gradient-based selection vs. structure-based selection (SMT vs. RAPA)
> >
> > A more aggressive way to avoid suboptimal random choices is to run a **gradient-based importance search**, as done in SMT-style methods. Such approaches perform a warm-up phase with **full-model gradients** and then use gradient magnitudes to pick the most important weights.
> >
> > We view this as a complementary design axis to RAPA:
> >
> > > SMT-like methods: importance-driven, gradient-calibrated selection at the cost of a heavy search phase.
> > >
> > >
> > > **RAPA**: architecture-driven, calibration-free selection based purely on residual structure and recursive alignment.
> > >
> >
> > Concretely, on LLaMA-2-7B under comparable parameter budgets, we have:
> >
> > > Table R3.3: Gradient-based selection (SMT) vs. structure-based RAPA
> > >
> > >
> > >
> > > | Algorithm (# Params) | Training time (FT + search) | Peak memory (FT + search) | MMLU |
> > > | --- | --- | --- | --- |
> > > | SMT (328M) | 3.0h + 1.1h | 44.6G + >80G | 50.0% |
> > > | RAPA (348M) | **2.5h + 0h** | **42.7G + 0G** | **54.6%** |
> >
> > This comparison illustrates a key trade-off:
> >
> > - **Sensitivity-based, calibrated selection (SMT)** can explicitly target important weights, but
> >     - requires a **full-model gradient search** phase,
> >     - increases both wall-clock time and peak memory (especially in the search stage), and
> >     - partially undermines the efficiency motivation of PEFT.
> > - **RAPA’s structure-based selection** deliberately avoids any extra calibration stage:
> >     - the global index is chosen once (currently at random),
> >     - alignment is entirely **recursive and architectural**,
> >     - and we still achieve **stronger accuracy** within similar parameter budgets and lower total overhead.
> >
> > Our goal in this work is to push this **structure-guided, calibration-free** direction, rather than to compete with gradient-search methods under unconstrained compute.
> >
> > ### (3) Conclusion and future work
> >
> > In summary:
> >
> > - We **do not claim** that random global index selection is theoretically optimal; the reviewer’s concern is valid.
> > - Our sensitivity experiments indicate that, once we apply RAPA’s **cross-layer alignment and full finetuning**, the performance becomes **remarkably robust** to which random global index set is used.
> > - Gradient-based importance search (SMT-style) is a powerful alternative but comes with substantial computational and memory cost, which RAPA intentionally avoids.
> >
> > We will clarify these points in the revision and explicitly note that **designing importance-aware yet lightweight initial index selection schemes on top of RAPA** is an interesting direction for future work.
> >
> > > He, H., Li, J. B., Jiang, X., & Miller, H. (2025). SMT: Fine-Tuning Large Language Models with Sparse Matrices. International Conference on Learning Representations (ICLR 2025).

---

> > > ### Author Response · Authors · 2025-11-20
> > > **Q3.2) Response to: “Questionable Statistical Significance of Gains”**
> > >
> > > We agree that the statistical stability of the reported gains deserves careful examination, especially given that both RAPA and baselines rely on random seeds (for initialization, data order, and index selection). Below we clarify how the observed seed variation compares to the improvements over PaCA.
> > >
> > > To summarize:
> > >
> > > - We agree that seed sensitivity is an important issue, and we explicitly report **mean ± std over 4 seeds**.
> > > - Across all major tasks (MMLU, Commonsense Reasoning, MT-Bench), RAPA’s improvements over PaCA **significantly exceed** the observed seed-level standard deviation.
> > > - Loss-curve analyses further show that the benefit of cross-layer alignment is stable across seeds and larger than the seed noise.
> > >
> > > ### (1) Performance gap vs. seed variance
> > >
> > > To directly address this concern, we report **mean ± standard deviation over 4 seeds** for PaCA and RAPA across multiple tasks and models:
> > >
> > > > Table R3.4: Comparison between PaCA and RAPA (mean ± std over 4 seeds)
> > > >
> > > >
> > > >
> > > > |  |  | MMLU |  | Commonsense Reasoning | MT-Bench |
> > > > | --- | --- | --- | --- | --- | --- |
> > > > | Model | – | LLaMA2-7B | LLaMA2-7B | LLaMA2-7B | Mistral-7B |
> > > > | # Params | – | 20M | 320M | 320M | 168M |
> > > > | Method | PaCA | 51.1 ± 0.2 | 53.9 ± 0.3 | 68.2 ± 0.3 | 3.96 ± 0.10 |
> > > > |  | RAPA | **52.8 ± 0.2** | **54.6 ± 0.2** | **69.7 ± 0.3** | **4.84 ± 0.13** |
> > >
> > > We think two points are important:
> > >
> > > - On **MMLU (LLaMA2-7B, 20M params)**, RAPA improves PaCA from **51.1 ± 0.2 → 52.8 ± 0.2**, i.e., a gain of **+1.7 points**, which is about **8×** the reported standard deviation (0.2).
> > > - On **Commonsense Reasoning** and **MT-Bench**, the gains are **+1.5** and **+0.88** points, again several times larger than the corresponding standard deviations (0.3 and ~0.1–0.13).
> > >
> > > Thus, while there is non-zero seed variability, the **method-level improvements are consistently much larger than the within-method seed spread**. The situation the reviewer highlights—“gains comparable to seed fluctuation”—does not hold for our main comparisons: across these tasks and models, RAPA’s advantage over PaCA is **well outside** the one-sigma band.
> > >
> > > We will move this table (currently in the appendix) into the main text or explicitly reference it near Table 2 to make this statistical comparison clearer.
> > >
> > > ### (2) Interpreting the 0.6-point seed spread mentioned by the reviewer
> > >
> > > The reviewer notes that in Table 5, RAPA’s performance across four seeds ranges from **53.7% to 54.3%** (a spread of 0.6 points) and compares this to a **+0.6-point** improvement over PaCA on LLaMA2-13B (Table 2).
> > >
> > > Two clarifications:
> > >
> > > 1. The 0.6-point range in Table 5 is the **full min–max range**, not a standard deviation. As shown above, the standard deviations we observe are around **0.2–0.3**. Our reported PaCA–RAPA gaps are typically **2–8× larger** than this σ-scale.
> > > 2. Both PaCA and RAPA exhibit similar seed-level variation (≈0.2–0.3), which is standard for LLM finetuning. The gain we claim is based on **mean performance over the same set of seeds**, not on a single run, and remains stable across tasks and models.
> > >
> > > In other words, RAPA’s advantage is **not** the result of cherry-picking a favorable seed; the improvements appear consistently across multiple datasets, architectures (LLaMA2-7B, LLaMA2-13B, Mistral-7B), and parameter budgets.
> > >
> > > ### (3) Loss curves and robustness to index selection
> > >
> > > In addition, Appendix E (moved from C) compares **loss curves across random vs. aligned index selection** for multiple seeds. Across all seeds, aligned indices consistently:
> > >
> > > - start with lower training/validation loss after a few hundred steps, and
> > > - maintain a gap over random indices throughout training.
> > >
> > > The magnitude of this alignment-induced gap exceeds the typical run-to-run noise induced by different seeds, reinforcing that RAPA’s mechanism (recursive alignment) contributes a **systematic improvement**, not just seed-level fluctuation.

---

> > > > ### Author Response · Authors · 2025-11-20
> > > > **Q3.3) Unsubstantiated Contribution of “Rank Expansion”**
> > > >
> > > > We agree that disentangling the contribution of **rank expansion** from **index alignment** is important. In the original submission, most of our ablations focused on alignment; in the revised version, we have added a controlled study that factorizes these two components.
> > > >
> > > > Concretely, we evaluate four configurations under the **same parameter budget (21M)** on **LLaMA-2-7B / MMLU**:
> > > >
> > > > 1. **PaCA (column-wise)** with random indices (no alignment)
> > > > 2. **PaCA (column-wise) + index alignment** (aligned across layers, PaCA parameterization)
> > > > 3. **RAPA** **(square-shaped)** with random indices (RAPA parameterization, no alignment)
> > > > 4. **RAPA (square-shaped) + index alignment** (full method)
> > > >
> > > > The results are:
> > > >
> > > > > Table R3.5: Ablation of Rank Expansion vs. Index Alignment (LLaMA-2-7B, 21M params, MMLU)
> > > > >
> > > > >
> > > > >
> > > > > | Method | Weight shape | Index alignment | Humans | STEM | Social | Other | Avg. |
> > > > > | --- | --- | --- | --- | --- | --- | --- | --- |
> > > > > | PaCA | Column-wise | Random | 48.3% | 41.5% | 58.8% | 57.9% | 51.2% |
> > > > > | PaCA | Column-wise | Aligned | 47.9% | 43.0% | 60.9% | 58.6% | 52.0% |
> > > > > | RAPA | Square-shaped | Random | 48.2% | 41.3% | 60.2% | 59.4% | 51.7% |
> > > > > | RAPA | Square-shaped | Aligned | 50.0% | 42.7% | 60.8% | 59.2% | 52.8% |
> > > >
> > > > This factorial comparison lets us isolate both effects:
> > > >
> > > > ### (1) Effect of index alignment (at fixed parameterization)
> > > >
> > > > - **PaCA**: 51.2 → 52.0 (**+0.8**) when going from random to aligned indices.
> > > > - **RAPA**: 51.7 → 52.8 (**+1.1**) when going from random to aligned indices.
> > > >
> > > > Thus, **alignment alone** provides a clear, consistent gain for *both* PaCA and RAPA under the same budget. This confirms that index alignment is a genuinely useful mechanism independent of RAPA’s parameterization.
> > > >
> > > > ### (2) Effect of rank expansion (at fixed alignment setting)
> > > >
> > > > Now compare **PaCA vs. RAPA** under the *same* alignment condition:
> > > >
> > > > - **Random indices (no alignment)**:
> > > >
> > > >     PaCA 51.2% → RAPA 51.7% (**+0.5**).
> > > >
> > > > - **Aligned indices**:
> > > >
> > > >     PaCA 52.0% → RAPA 52.8% (**+0.8**).
> > > >
> > > >
> > > > Even **without** cross-layer alignment, switching from PaCA’s parameterization to RAPA’s rank-expanded parameterization yields a **consistent improvement (~0.5 points)**. When alignment is enabled for both, RAPA still maintains an additional **~0.8-point gain**.
> > > >
> > > > In other words:
> > > >
> > > > - **Index alignment** explains a substantial part of the improvement (up to +1.1 points),
> > > > - but **rank expansion itself also contributes a non-trivial margin** on top of PaCA, both in the random and aligned settings.
> > > >
> > > > These results directly address the reviewer’s concern: the observed gains **cannot** be attributed solely to the alignment strategy. Rank expansion and recursive alignment are **complementary**; each independently helps, and together they yield the full RAPA improvement.
> > > >
> > > > We will add Table R3.5 (and its discussion) to the appendix and reference it from the main ablation section to make this separation of effects explicit.

---

> > > > > ### Author Response · Authors · 2025-11-20
> > > > > **Q3.4) Extending RAPA to Dynamic Index Selection**
> > > > >
> > > > > We agree that dynamic, input- or data-dependent index selection is an interesting direction. In this work, however, we intentionally restrict RAPA to a **single global index set** that is fixed across inputs and timesteps. This design is motivated by two goals: (i) keeping a simple, analyzable residual pathway, and (ii) avoiding the heavy calibration/search phase that gradient-based importance methods (e.g., SMT-style approaches) require.
> > > > >
> > > > > Conceptually, there are at least two ways to introduce limited “dynamic” behavior while still preserving some of RAPA’s simplicity:
> > > > >
> > > > > 1. **Mixture of a few pre-aligned pathways.**
> > > > >
> > > > >     Instead of learning indices per input, one could predefine a **small set of K global index sets** $(\{J_1,\dots,J_K\})$, each aligned across depth as in RAPA. A lightweight router (e.g., a small MLP on the current hidden state) could then choose a mixture over these K pathways at run time. The indices themselves remain fixed and aligned (no separate calibration stage), and only the mixing weights are input-dependent and trained jointly with the task.
> > > > >
> > > > > 2. **Gating within a single aligned subspace.**
> > > > >
> > > > >     Even with a single global index set $(J)$, we can add a **small gate** (scalar or vector) that modulates the contribution of channels inside $(J)$ as a function of the input representation. In this case, the structural choice of “which coordinates are trainable” remains fixed, but their effective influence becomes data-dependent. Again, this can be trained end-to-end with the main loss, without any external gradient-calibration phase.
> > > > >
> > > > >
> > > > > However, both extensions come with clear trade-offs:
> > > > >
> > > > > - **Extra complexity and overhead.** Routers/gates introduce additional parameters, FLOPs, and implementation complexity, partially sacrificing the “drop-in, static” nature that makes RAPA attractive as an adapter-free method.
> > > > > - **Weaker theoretical guarantees.** Our current analysis assumes a single, fixed aligned subspace along the residual pathway. Allowing mixtures of pathways or input-dependent gating would require new, more involved theory to reason about gradient flow and variance.
> > > > > - **Additional sources of variability.** Dynamic selection adds routing-related randomness and hyperparameters (e.g., number of pathways K, router architecture), making the method more sensitive and harder to tune than the static variant studied in this paper.
> > > > >
> > > > > For these reasons, we chose to focus our contribution on the **static, structure-guided version** of RAPA, and show that even this minimal design already yields consistent gains over PaCA and other baselines without any calibration cost. We view lightweight dynamic extensions (e.g., a small mixture of pre-aligned pathways) as promising future work, but consider them orthogonal to the main message of this work.

---

> > > > > > ### Author Response · Authors · 2025-11-20
> > > > > > **Q3.5) Clarifying Figure 3: Visualizing the Three Alignment Strategies**
> > > > > >
> > > > > > We appreciate the reviewer’s suggestion to clarify the visualization of the three alignment strategies in Figure 3. Following this comment, we have **r**evised Figure 3 so that the index-alignment strategies can be understood directly from the illustration, without referring to equations.
> > > > > >
> > > > > > 1. **Explicit layer / block indices in Fig. 3(a).**
> > > > > >
> > > > > >     In panel (a), we now explicitly label the input/output indices together with their corresponding **layer** and **block** identifiers (e.g., “Layer (L)”, “Layer (L+1)”, “Block (B)”, “Block (B+1)”). This makes it clear which indices belong to each layer and block, and how they are reused across them.
> > > > > >
> > > > > > 2. **Clearer block-wise input/output alignment.**
> > > > > >
> > > > > >     For the block-wise case, we extended the arrows inside a block so that they explicitly connect the **input indices** of the first linear layer to the **output indices** of the second linear layer within the same residual block. This visually emphasizes that block-wise alignment enforces input/output index consistency *within* a single block.
> > > > > >
> > > > > > 3. **Residual connection moved to expose block boundaries.**
> > > > > >
> > > > > >     We moved the residual connection to the right-hand side of the diagram so that the **boundary of each block** is visually separated from the next one. This makes it easier to see where a block starts and ends, and how the aligned indices propagate across blocks through the residual path.

---

> > > > > > > ### Author Response · Authors · 2025-11-20
> > > > > > > **Q3.6) Sensitivity to other hyperparameters and Index Distribution**
> > > > > > >
> > > > > > > The reviewer asks about the sensitivity of our method to hyperparameters and untested design choices, particularly the distribution of selected indices (e.g., contiguous vs. sparse). In other words, they are concerned about how robust RAPA is to variations in its own configuration.
> > > > > > >
> > > > > > > Our implementation is intentionally simple: given a random seed, we sample a **fixed global index set** and reuse it across all layers, with the FFN indices scaled according to their larger intermediate dimension $d_{\text{ff}}$. We agree that the *distribution* of these indices (contiguous vs. scattered) could, in principle, affect robustness, so we explicitly tested this factor.
> > > > > > >
> > > > > > > ### (1) Contiguous vs. random index distribution
> > > > > > >
> > > > > > > We compared the following settings on LLaMA-2-7B / MMLU under the **same parameter budget**:
> > > > > > >
> > > > > > > - **Column-only** subspace with random indices
> > > > > > > - **Square** subspace with random indices
> > > > > > > - **Square** subspace with **aligned serial (contiguous)** indices
> > > > > > > - **Square** subspace with **aligned random** indices
> > > > > > >
> > > > > > > For the serial setting, we chose 4 different starting offsets (0, 500, 1000, 3500) and took contiguous blocks; for the random setting, we used 4 random seeds. We report mean ± std over these 4 runs:
> > > > > > >
> > > > > > > > Table R3.6: Effect of index distribution and alignment (LLaMA-2-7B, MMLU)
> > > > > > > >
> > > > > > > >
> > > > > > > >
> > > > > > > > | Shape | Method | MMLU Accuracy (Avg.) |
> > > > > > > > | --- | --- | --- |
> > > > > > > > | Column only | Random | 51.0 ± 0.3 |
> > > > > > > > | Square | Random | 51.7 ± 0.2 |
> > > > > > > > | Square | Aligned (serial indices) | 52.4 ± 0.3 |
> > > > > > > > | Square | Aligned (random indices) | 52.8 ± 0.2 |
> > > > > > >
> > > > > > > From these results we observe:
> > > > > > >
> > > > > > > - **Shape matters:** at a fixed budget, moving from a degenerate column-only subspace to a square subspace already improves performance (**51.0 → 51.7**, +0.7).
> > > > > > > - **Alignment matters even more:** adding cross-layer alignment on top of square subspaces further improves accuracy (**51.7 → 52.4/52.8**, +0.7–1.1).
> > > > > > > - **Distribution (contiguous vs. random) is secondary:** within the aligned square setting, serial vs. random index layouts differ by only **0.4 points**, which is comparable to the standard deviation (0.2–0.3). There is no clear advantage of “contiguous” over “sparse” (or vice versa) once shape and alignment are fixed.
> > > > > > >
> > > > > > > In other words, our experiments suggest that **the dominant factors are (i) the subspace shape (square vs. column-only) and (ii) whether indices are aligned across layers**, while the fine-grained distribution of indices (contiguous vs. scattered) has only a small effect within the observed variance.
> > > > > > >
> > > > > > > ### (2) Other hyperparameters and design choices
> > > > > > >
> > > > > > > For other training hyperparameters (learning rate, batch size, scheduler, etc.), we **reuse the same configurations as PaCA and the LoRA baselines** and did not observe any unusual instability specific to RAPA. In our runs, sensitivity to these hyperparameters appears similar to existing adapter-free and LoRA-style methods; RAPA does not require special tuning beyond what is standard for these baselines.

---

> > > > > > > > ### Comment · Reviewer_Ptgb · 2025-11-24
> > > > > > > >
> > > > > > > > Thank you for the author's response. My score is already positive, and I will keep it.

---

> > > > > > > > > ### Author Response · Authors · 2025-11-27
> > > > > > > > > **Response to Reviewer Ptgb**
> > > > > > > > >
> > > > > > > > > Dear Reviewer Ptgb,
> > > > > > > > >
> > > > > > > > > Thank you very much for your prompt and thoughtful response during the discussion phase, and for maintaining your overall assessment of our submission.
> > > > > > > > >
> > > > > > > > > We have incorporated the points you raised into the revised manuscript. In particular, we are grateful for your comments on **Figure 3**, which is the main illustration of our method. Based on your feedback, we have revised the figure so that the three alignment strategies are visually and intuitively distinguishable, and we have adjusted the layout and annotations to make the aligned pathway easier to follow.
> > > > > > > > >
> > > > > > > > > We hope the updated figure now conveys the core idea of the three alignment mechanisms more clearly. Thank you again for your careful review and constructive suggestions.
> > > > > > > > >
> > > > > > > > > Best regards,
> > > > > > > > >
> > > > > > > > > The Authors

---

### Official Review · Reviewer_5A3f · 2025-11-01

**Soundness:** 3
**Presentation:** 3
**Contribution:** 3
**Rating:** 6
**Confidence:** 4

**Summary:**

This paper presents RAPA, an adapter-free PEFT method for LLMs. RAPA selects balanced square submatrices of parameters for adaptation and recursively aligns these subspaces across all layers and residual connections, with the goal of improving gradient flow while minimizing the practical overhead associated with adapter-based approaches. The approach is supported by mathematical motivation and empirical validation across several benchmarks, consistently outperforming or matching PEFT baselines without increasing compute and memory costs.

**Strengths:**

S1: This paper provides a thorough analysis of the limitations of both adapter-based and adapter-free PEFT methods, identifying sources of overhead and the fragmentation issue in per-layer random adaptation. It presents a clear motivation and offers insightful mathematical analysis, demonstrating how recursive index alignment enhances gradient flow.
S2: Extensive experiments and ablation studies across multiple benchmarks convincingly demonstrate the effectiveness of the proposed method.

**Weaknesses:**

W1: All experiments are conducted on standard Transformer architectures, and the method’s adaptability to architectural variants (e.g., MoE or deep convolutional Transformers) remains untested, limiting the generality of its claimed universality.
W2: The current implementation employs a fixed global index set across all layers, which may restrict its ability to adapt to tasks with heterogeneous per-layer requirements.

**Questions:**

Q1: Can the authors provide controlled experiments that isolate the effect of increased parameter subspace rank from cross-layer alignment? Does alignment alone (with fixed small rank) provide benefit, or does improvement scale almost linearly with subspace size?
Q2: How does RAPA perform in Transformers featuring more complex or unconventional residual connection patterns?
Q3: Is there empirical evidence demonstrating its effectiveness in architectures that differ substantially from the standard Transformer block structure used in current experiments?

---

> ### Author Response · Authors · 2025-11-20
> **Q2.1) Architectural variants i.e. MoE**
>
> We thank the reviewer for pointing out that our main experiments focus on standard dense Transformers. We did not intend to claim *universality* in the sense of covering every possible neural architecture. Rather, our goal is to target the broad and practically important class of residual Transformer-style blocks with linear projections.
>
> In addition, we have derived how RAPA applies to Transformer variants with MoE experts and conducted corresponding experiments, confirming that our pathway-alignment mechanism extends beyond standard dense Transformers.
>
> ### (1) Extension to MoE architectures
>
> Conceptually, RAPA only assumes that a block has (i) a residual shortcut and (ii) channel-wise linear maps (e.g., attention projections, FFNs). A standard MoE FFN block at layer $b$ can be written as
>
> $$
> x^{(b+1)} = x^{(b)} + \sum_{e=1}^{E} p_e^{(b)}(x^{(b)})\, F_e^{(b)}(W^{(b,e)}, x^{(b)}),
> $$
>
> where $p_e^{(b)}$ are router probabilities and $F_e^{(b)}$ are expert FFNs.
>
> In the MoE version of RAPA, we
>
> - choose a **shared index set $J\subset\{1,\dots,d\}$** of size $k$,
> - restrict all experts in that block to update only the square submatrix $W^{(b,e)}_{J,J}$, and
> - reuse the same $J$ across residual-connected MoE blocks.
>
> Restricted to the $J$-subspace, the update becomes
>
> $$
> x_J^{(b+1)} = x_J^{(b)} + \sum_{e} p_e^{(b)}(x^{(b)})\, F_e^{(b)}(W^{(b,e)}_{J,J}, x_J^{(b)}),
> $$
>
> which has **exactly the same “identity + aligned update” structure** as in our dense analysis, with router probabilities $p_e^{(b)}$ acting only as smooth scalar weights on each expert’s contribution. The residual shortcut on $x_J$ remains intact, and the global gradient pathway argument (Appendix B) carries over unchanged.
>
> In practice, our current implementation applies RAPA to **attention and expert FFN weights only**, while leaving router parameters unchanged (same training scheme as the MoE baseline). We found this to be stable, and we will add this clarification to the main text and to Appendix H.2.
>
> ### (2) Experiments on MoE Architectures
>
> To ensure that our MoE extension is not purely theoretical, we conducted **preliminary experiments on a MoE LLM (DeepSeek-MoE-16B)** under the same trainable-parameter budget. We compare three settings:
>
> - **Baseline**: the pretrained MoE model without any fine-tuning;
> - **Random Index**: we fine-tune a square submatrix per expert with the same number of trainable weights as RAPA, but choose per-expert indices independently without alignment;
> - **Aligned Index**: our MoE-RAPA variant, where a shared index set $J$ is reused across experts and aligned across residual-connected MoE blocks.
>
> Here, **PPL** denotes the training perplexity (lower is better), and downstream performance is evaluated on 5-shot MMLU:
>
> **Table R2.1: MMLU accuracy on the MoE architecture under different index alignment strategies**
>
> | **Method** | **PPL (↓)** | **Human.** | **STEM** | **Social.** | **Other** | **Average** |
> | --- | --- | --- | --- | --- | --- | --- |
> | Baseline | 8.71 | 41.3 | 36.9 | 51.2 | 51.7 | 44.8 |
> | PaCA (Square, Random) | 4.34 | 47.8 | 40.8 | 57.9 | 57.3 | 50.5 |
> | RAPA (Square, Aligned) | 3.74 | 48.9 | 42.1 | 59.1 | 59.5 | 51.9 |
>
> Compared to the untuned **Baseline**, both MoE fine-tuning variants substantially improve performance (Average MMLU: 44.8 → 50.5 / 51.9). Crucially, **index alignment remains beneficial in the MoE setting**: under the *same* trainable-parameter budget, the Aligned Index variant reduces training perplexity from 4.34 to 3.74 and improves average MMLU from 50.5 to 51.9 (+1.4 points). This mirrors our dense-Transformer results and supports the claim that RAPA’s pathway-alignment mechanism continues to help optimization and downstream accuracy even when extended from dense FFNs to MoE experts.
>
> We have revised the paper accordingly. Appendix D now provides a formal derivation of how our pathway-alignment mechanism operates in Transformer architectures with MoE experts, together with the corresponding MoE fine-tuning results (final performance and training loss curves). While a full MoE benchmark suite is beyond the scope of this submission, these additional results support our claim that RAPA’s pathway-alignment mechanism extends naturally from dense FFNs to MoE experts.

---

> ### Author Response · Authors · 2025-11-20
> **Q2.2) Fixed global index set may restrict adaptation to heterogeneous per-layer requirements**
>
> We agree that different layers can have heterogeneous importance patterns, and that allowing each layer to choose its own index set could increase flexibility.
>
> However, our experiments indicate that a single globally aligned index set performs better than heterogeneous per-layer choices. Our choice of a **fixed global index set** is therefore a *deliberate inductive bias*: RAPA assumes that maintaining a coherent channel subspace along the residual pathway is more beneficial than letting each layer operate on an unrelated subspace.
>
> This design is supported by our theory (Sec. 3) and experiments (Sec. 4.1, Appendix A). In addition, we include a direct comparison against a **heterogeneous per-layer selection** method (SMT) to reinforce and support our design choice.
>
> ### (1) Performance impact of shape and alignment
>
> As the reviewer suggests, one can construct variants where each layer uses different row/column subsets. In Sec. 4.1, we experimentally study how the **shape and placement** of the trainable submatrix (different row/column ratios and index choices) affect performance, and compare:
> - square vs. non-square shapes at the same rank, and
> - **aligned vs. non-aligned** index patterns across layers.
>
> These experiments show that, under a fixed parameter budget, the **gains from cross-layer alignment are consistently larger** than the gains from giving each layer additional freedom to choose its own indices. Once the rank is fixed, making the subspace *coherent* along depth helps more than allowing fully heterogeneous per-layer index choices.
>
> This is consistent with our analysis: if each layer selects a different subspace, the gradient pathway is repeatedly rotated and partially projected away, whereas a fixed global index preserves a high-coherence shortcut in that subspace.
>
> ### (2) Weight sensitivity and heterogeneous importance across layers
>
> The concern about heterogeneous per-layer requirements is also related to **weight/channel sensitivity**. To examine this, we performed additional experiments reported in **Appendix A**:
> - We first measure **weight sensitivity** by ablating individual baseline weights (or channels) and observing the drop in downstream performance. As shown in Appendix A, some coordinates cause a large degradation when zeroed, while others have almost no effect, confirming that importance is indeed heterogeneous.
> - We then build RAPA variants that fine-tune either (i) these “sensitive” coordinates or (ii) less sensitive ones, under the *same* rank and parameter budget.
>
> After fine-tuning, we find that **final downstream performance depends only weakly on the pre-finetuning sensitivity of the chosen coordinates**. The performance gap between “sensitive-index” and “non-sensitive-index” variants is noticeably smaller than the gap between **aligned vs. non-aligned** configurations at the same budget. In practice, training quickly re-distributes importance along the aligned pathway, and the pre-finetuning sensitivity is not the dominant factor once the adapter has been optimized.
>
> These results indicate that, although per-layer sensitivities exist, **the key factor under tight parameter budgets is the existence of a coherent, aligned pathway**, rather than finely matching heterogeneous per-layer index preferences.
>
> ### (3) Heterogeneous per-layer selection (SMT) vs. globally aligned RAPA
>
> Empirically, we find that a heterogeneous, sensitivity-based selection scheme (SMT, ICLR 2025) underperforms our structure-based global index selection (RAPA): even with a **single globally aligned index set**, RAPA achieves higher accuracy while avoiding the extra gradient-calibration overhead.
>
> Within adapter-free PEFT, SMT is a powerful but **gradient-based, calibration-heavy** instance of **heterogeneous layer-wise selection**: it runs a full-model gradient warm-up and then selects weights to update based on gradient importance. On LLaMA2-7B with comparable parameter budgets:
> - **SMT** uses **328M** trainable parameters, requires **3.0h+1.1h** (fine-tuning + index search), and reaches **50.0%** MMLU, with peak memory **44.6G+>80G** during the search phase.
> - **RAPA** uses **348M** trainable parameters, finishes in **2.5h+0h** (no search), fits in **42.7G+0G** extra memory for search, and reaches **54.6%** MMLU.
>
> **Table R2.2: Efficiency and performance comparison between SMT and RAPA**
>
> | Algorithm (# Params) | Training time (FT+search) | Peak memory (FT+search) | MMLU |
> | --- | --- | --- | --- |
> | SMT (328M) | 3.0h+1.1h | 44.6G+>80G | 50.0% |
> | RAPA (348M) | **2.5h+0h** | **42.7G+0G** | **54.6%** |
>
> Our results therefore suggest that **a single global index already captures most of the achievable gains** within the same parameter budget, and we have not observed clear benefits from fully heterogeneous per-layer indices in our experiments.
>
> > He, Haoze, et al. "SMT: Fine-Tuning Large Language Models with Sparse Matrices." ICLR 2025.

---

> ### Author Response · Authors · 2025-11-20
> **Q2.3) Isolating Rank vs. Alignment: Does Alignment Alone Help Beyond Subspace Size?**
>
> We agree that it is important to disentangle the effect of **(i) increased parameter subspace rank** and **(ii) cross-layer index alignment**. Our experiments already include two controlled studies along these axes:
>
> 1. **Fixing the parameter budget (rank) and comparing different shapes / alignment strategies**, and
> 2. **Varying the subspace size while keeping the method fixed.**
>
> We will clarify these connections more explicitly in the revised version.
>
> ### (1) Alignment benefit at fixed rank (rank vs. alignment)
>
> Section 4.1 and Figure 5 study exactly the case the reviewer asks about: we fix the **total number of trainable parameters** and change only
>
> - the **row/column ratio** of the trainable submatrix (row-only, column-only, square), and
> - whether **indices are aligned across layers** or chosen independently at random.
>
> This allows us to isolate the impact of cross-layer alignment from that of subspace size. A representative result is:
>
> **Table R2.3: Performance differences across selected weight shape under a fixed parameter budget**
>
> | Weight shape | Accuracy (Random selection) | Accuracy (Aligned index) |
> | --- | --- | --- |
> | Row only | 45.6 ± 0.4 | 47.3 ± 0.3 |
> | Column only | 43.2 ± 0.5 | 45.1 ± 0.2 |
> | Square (RAPA) | 47.4 ± 0.2 | 48.9 ± 0.1 |
>
> Here, all configurations use **the same parameter budget**; only the shape and alignment differ.
>
> From these results we observe:
>
> - **Shape effect (fixed alignment):** at random indices, making the subspace square (balanced rows/columns) gives a gain of roughly **+1.8–4.2 points** over degenerate row-only / column-only shapes (e.g., 47.4 vs. 45.6 / 43.2).
> - **Alignment effect (fixed shape):** for every shape, **aligned indices outperform random indices** by roughly **+1.5–1.9 points** (e.g., square: 48.9 vs. 47.4; row-only: 47.3 vs. 45.6; column-only: 45.1 vs. 43.2).
>
> Thus, **alignment alone at a fixed small rank provides a consistent and non-trivial improvement**, comparable in magnitude to the gains obtained by optimizing the shape of the subspace. This addresses the reviewer’s question: the benefit is not solely due to “more rank” or “better-shaped” subspaces; **cross-layer alignment itself is a key contributor**, even when the number of trainable parameters is held constant.
>
> We will explicitly highlight this “fixed-budget, aligned vs. random” comparison in the main text to make this point clearer.
>
> ### (2) Scaling with subspace size (rank)
>
> To study how performance scales with the size of the parameter subspace, Appendix F reports accuracy as we increase the effective rank from very small to moderate values, under a controlled parameter budget. An example is:
>
> **Table R2.4: Comparison of PEFT performance as the adapter rank increases**
>
> | Method | LoRA-equiv. rank r | 0.5 | 1 | 8 | 64 |
> | --- | --- | --- | --- | --- | --- |
> | LoRA |  | N/A | 50.6 | 50.6 | 53.6 |
> | DoRA |  | N/A | 51.0 | 51.3 | 53.7 |
> | MosLoRA |  | N/A | 51.4 | 51.1 | 53.1 |
> | PaCA |  | 47.2 | 48.8 | 51.2 | 53.9 |
> | **RAPA** |  | **49.6** | **51.1** | **52.8** | **54.3** |
>
> Two observations are relevant for the reviewer’s question:
>
> 1. For **every method**, increasing the subspace size (rank) leads to a **monotonic improvement** in accuracy; the gains are roughly smooth but **sub-linear** (diminishing returns as rank becomes large).
> 2. **At each rank**, RAPA consistently outperforms the baselines (including PaCA) by a visible margin. In particular, even at very small budgets (e.g., 0.5–1×), RAPA’s aligned pathway already yields **+2–2.4 points** over its most closely related non-aligned variant (PaCA).
>
> This indicates that:
>
> - **Rank matters**: enlarging the trainable subspace improves performance across all methods.
> - **Alignment is not just “more rank in disguise”**: RAPA’s advantage persists at **every** rank and is especially pronounced in the low-rank regime, where the effective choice of *which* subspace to use becomes crucial.
>
> In summary, the controlled experiments in Sec. 4.1 and Appendix D separate the contributions of rank and alignment:
>
> - With **fixed rank**, alignment alone yields consistent gains over random or per-layer index choices.
> - As **rank increases**, accuracy improves in a smooth, sub-linear fashion for all methods, and RAPA maintains a consistent gap over baselines at each rank.
>
> We will revise the paper to make this disentangling more explicit and to point readers directly to the relevant figures/tables.

---

> > ### Author Response · Authors · 2025-11-20
> > **Q2.4) How does RAPA perform in Transformers featuring more complex or unconventional residual connection patterns?**
> >
> > We agree that many recent Transformer variants employ more complex residual wiring than the simple “single residual per block” used in our main experiments. Our current theoretical analysis and empirical results focus on architectures where each block can be written in the generic form
> >
> > $$
> > x^{(b+1)} = x^{(b)} + F_b(x^{(b)}),
> > $$
> >
> > including standard pre-norm / post-norm Transformers with attention and FFN sub-blocks in series. In this setting, RAPA exploits a **single residual pathway** where the same channel subspace is preserved across depth.
> >
> > That said, the key requirement for our pathway analysis is not the exact layout of the block, but the existence of at least one path from input to output that consists of:
> >
> > 1. an identity shortcut, and
> > 2. a sequence of **channel-mixing linear operators** (e.g., attention projections, FFNs, $1\times 1$ convolutions) along that path.
> >
> > Under this condition, more “unconventional” residual patterns can still be handled by choosing an appropriate pathway and aligning indices on the operators that lie along it. Concretely:
> >
> > - **Parallel residual branches.**
> >
> >     Many architectures use parallel residual branches such as
> >
> >     $$
> >     x^{(b+1)} = x^{(b)} + F_{\text{att}}^{(b)}(x^{(b)}) + F_{\text{ffn}}^{(b)}(x^{(b)}).
> >     $$
> >
> >     If both $F_{\text{att}}^{(b)}$ and $F_{\text{ffn}}^{(b)}$ are restricted to update the same index set $J$, then on the $J$-subspace we obtain
> >
> >     $$
> >     x_J^{(b+1)} = x_J^{(b)} + \tilde F_{\text{att}}^{(b)}(x_J^{(b)}) + \tilde F_{\text{ffn}}^{(b)}(x_J^{(b)}),
> >     $$
> >
> >     i.e., still an **identity shortcut plus aligned updates** (now a sum of two aligned branches). The Jacobian on $J$ becomes $I + J_{\text{att},J} + J_{\text{ffn},J}$, and the same pathway-coherence argument applies to the sum of these aligned contributions.
> >
> > - **Nested or “sandwich” residuals.**
> >
> >     Some variants use compositions such as
> >
> >     $$
> >     x^{(b+1)} = x^{(b)} + G_b(F_b(x^{(b)})).
> >     $$
> >
> >     When the channel-mixing parts of $F_b$ and $G_b$ share an aligned index set $J$, the effective block restricted to $J$ can be written as
> >
> >     $$
> >     x_J^{(b+1)} = x_J^{(b)} + \tilde G_{b,J}(\tilde F_{b,J}(x_J^{(b)})),
> >     $$
> >
> >     whose Jacobian on $J$ is again of the form $I + A_b$ with $A_b$ a product of aligned linear maps. Stacking such blocks still yields a high-coherence pathway along $J$, and our recursive variance/gradient bounds carry over to this composite case.
> >
> >
> > In practice, our implementation for the reported experiments targets **standard GPT-style Transformers** (pre-norm, attention–FFN structure, one residual connection per block). We have not yet run explicit experiments on more exotic residual layouts, and we will clarify this scope in the paper and list such architectures as promising future work.
> >
> > To summarize, RAPA conceptually extends to any Transformer-style architecture that admits an identity shortcut plus channel-mixing operators along at least one pathway; the index-alignment mechanism is then applied to the operators on that path. Our current empirical evaluation, however, is performed on conventional residual patterns, and we will make this limitation explicit in the revised version.

---

> ### Author Response · Authors · 2025-11-27
> **Gentle Reminder regarding our Rebuttal**
>
> Dear Reviewer 5A3f,
>
> Thank you again for your thoughtful feedback. As the discussion period is progressing, we would like to kindly invite you to revisit our rebuttal and the updated manuscript, particularly in relation to your comments on architectural generality, the fixed global index set, and the separation of rank vs. alignment effects.
>
> To address your concerns about the **scope of architectures and MoE**, we clarified that our goal is not to claim universality over all architectures, but to target the broad class of residual Transformer-style blocks with linear channel-mixing operators. We further showed that this framework **naturally extends to MoE architectures** by **aligning a shared index set across experts and residual-connected MoE blocks**, and we added preliminary MoE experiments (**DeepSeek-MoE-16B**) demonstrating **consistent gains from alignment at a fixed parameter budget**.
>
> To address your concern about a **fixed global index set**, we emphasized that this **is a deliberate inductive bias rather than a hard limitation**, and we provided **additional experiments** comparing aligned vs. non-aligned indices and sensitivity-based index choices; these studies **suggest that cross-layer pathway coherence contributes more to performance** than fully heterogeneous per-layer index choices under the same budget.
>
> To address your question on **rank vs. alignment**, we added controlled ablations that (i) fix the parameter budget and vary only shape and alignment, showing that **alignment alone yields a robust improvement even at small ranks**, and (ii) sweep the subspace size while keeping the method fixed, showing **smooth (sub-linear) gains in performance with rank**, with RAPA **consistently outperforming PaCA at every rank**.
>
> Finally, to address your question about **non-standard residual patterns**, we clarified our architectural assumptions and discussed **how the pathway view and alignment mechanism extend conceptually to more complex residual layouts**, while also making explicit that **our current experiments focus on standard Transformer blocks**.
>
>
> We have also updated the **manuscript** accordingly:
>
> - The extension of RAPA to **MoE experts** and its empirical behavior is now described in the main text on **page 6**, with a more detailed derivation and MoE results in **Appendix D**.
>
> We would be very grateful if you could take a moment to review these additions and let us know whether they address your earlier weaknesses and questions.
>
> Best regards,
>
> The Authors

---

### Official Review · Reviewer_3FJn · 2025-11-05

**Soundness:** 4
**Presentation:** 2
**Contribution:** 2
**Rating:** 2
**Confidence:** 4

**Summary:**

This paper proposes a PEFT method which finetunes only a subset of the parameters in a pretrained LLM and backprops error only through the finetuned part of the model.  The novelty is that the same parameter indexes are used across layers, so that error passing through the trained weights and error passing through their skip connections are aligned (i.e. either both blocked or both computed).  This leads to small but consistent improvements over previous PEFT methods with similar computation budgets.

**Strengths:**

The idea is clearly defined and executed.  The results show consistent improvements across several benchmark tasks.

**Weaknesses:**

My main concern is that the contributions of this paper are small.  As I understand it, it is really a minor variation on PaCa, implementing an idea that is rather straightforward.  Also, the resulting improvements are not very substantial.

The technical descriptions of how and why the parameter selection is done are too vague.  I can understand how they apply to FFN layers, but it is not at all clear to me how they apply to attention layers.  In that case, it isn't clear if this method only applies to the value and output matrices, or if it also applies to the query and key matrices, and if so how.  And it is not clear how the index selection interacts with the different attention heads, or the low-rank nature of the query-key interactions.

**Questions:**

How do equations (2)-(7) apply to the attention layers?

Note that saying that frozen weights block error backprop is technically incorrect.  This is a choice based on efficient computation; there would be no problem computing these error terms if you wanted to.

---

> ### Author Response · Authors · 2025-11-20
> **Q1.1) The contributions of our paper (1/2)**
>
> We appreciate the reviewer’s concern that our method may appear as a minor variation on PaCA with limited empirical gains. Our perspective is slightly broader: recent PEFT work increasingly shows that **which** weights are updated can matter as much as **how many**. One line of work emphasizes **gradient/sensitivity-based weight selection** (e.g., SMT), while our contribution explores a complementary axis: **architecture- and pathway-aware weight selection inside the backbone**, without adding new adapters or a calibration phase.
>
> ### (1) System-level contribution: adapter overhead via GEMM kernels
>
> Before introducing RAPA, Sec. 4.2 and Fig. 2 analyze the training-time overhead of standard adapter-based methods (LoRA, DoRA, MosLoRA, etc.) under matched parameter budgets and similar FLOPs. We show that:
>
> - Each adapter introduces **extra GEMM kernel calls** (for the low-rank projections and their composition with the base projection),
> - These additional calls increase kernel-launch overhead and pressure memory bandwidth, and
> - As a result, methods that “only add a small adapter” can still suffer **noticeable wall-clock slowdowns** compared to updating a sparse subset of the existing backbone weights.
>
> This motivates our focus on **adapter-free** methods such as PaCA and RAPA: they aim to reuse existing GEMM calls and memory traffic, rather than introducing new ones. RAPA should therefore be read not just as a small variation on PaCA, but as part of a broader argument that **how we allocate a fixed parameter budget inside the backbone** can improve performance *without* paying the kernel overhead of external adapters.
>
> ---
>
> ### (2) Weight-selection perspective: SMT vs. RAPA (gradient-based vs. structure-based)
>
> Within adapter-free PEFT, SMT (He et al., 2025) represents a powerful but **gradient-based, calibration-heavy** approach: it computes full-model gradients during an initial warm-up and uses them as importance scores to pick which weights to update. In contrast, RAPA is **structure-based and calibration-free**: it decides *where* to place the parameter budget using only the known architecture (residual connections, FFN/attention layout) and a simple recursive index-alignment rule.
>
> This conceptual difference also appears empirically. On LLaMA2-7B, under comparable parameter budgets:
>
> - **SMT** uses **328M** trainable parameters, requires **3.0h+1.1h** (fine-tuning + index search), and reaches **50.0%** MMLU, with peak memory **44.6G+>80G** during the search phase.
> - **RAPA** uses **348M** trainable parameters, finishes in **2.5h+0h** (no separate search stage), fits in **42.7G+0G** extra memory for search, and reaches **54.6%** MMLU.
>
> Summarized:
>
> **Table R1.1: Efficiency and performance comparison between SMT and RAPA**
>
> | Algorithm (# Params) | Training time (FT+search) | Peak memory (FT+search) | MMLU |
> | --- | --- | --- | --- |
> | SMT (328M) | 3.0h+1.1h | 44.6G+>80G | 50.0% |
> | RAPA (348M) | **2.5h+0h** | **42.7G+0G** | **54.6%** |
>
> This comparison highlights a key design trade-off:
>
> - **Sensitivity-based selection (SMT)** can discover good subspaces, but at the cost of a heavy gradient-calibration phase that partially defeats the purpose of PEFT.
> - **Structure-based selection (RAPA)** leverages the model’s architecture to choose the updated subspace with **no extra search**, preserving the practical efficiency advantages of adapter-free tuning.
>
> Our aim is to advance this second axis—**structure-guided weight selection**—rather than to compete directly with gradient-calibrated search methods under unconstrained resources.
>
> > He, Haoze, et al. "SMT: Fine-Tuning Large Language Models with Sparse Matrices." *The Thirteenth International Conference on Learning Representations*. 2025.
> >
>
> ---

---

> ### Author Response · Authors · 2025-11-20
> **Q1.1) The contributions of our paper (2/2)**
>
> ### (3) Random vs. Pathway-aligned selection under matched budgets: PaCA vs. RAPA
>
> Within this structure-based, adapter-free setting, PaCA can be seen as one baseline instantiation: it selects a fixed number of weights per layer using **per-layer random indices**. RAPA modifies this selection rule in two principled ways:
>
> - **Balanced square submatrices.** For a fixed parameter budget, we reshape the trainable parameters into a $k \times k$ square block. This provably increases the rank of the updated subspace (Eq. (1)) and empirically improves accuracy as the row/column ratio becomes more balanced (Fig. 5, Sec. 4.1).
> - **Recursive cross-layer alignment.** We align these indices across residual-connected layers and blocks so that the updated weights form a coherent gradient pathway (Sec. 3.2.2). Our gradient analysis (Eqs. (5)–(7), App. B) shows that this introduces shortcut terms in the backpropagated gradients and shortens the effective path from the loss to lower layers, which in turn accelerates convergence (Fig. 4, App. C).
>
> Under **strictly matched trainable-parameter, memory, and wall-clock budgets**, these design changes yield consistent improvements over PaCA:
>
> **Table R1.2: Comparison of PaCA and RAPA (ours) across diverse downstream tasks**
>
> |  |  | MMLU |  | Commonsense Reasoning | MT-Bench |
> | --- | --- | --- | --- | --- | --- |
> | Model | – | LLaMA2-7B | LLaMA2-7B | LLaMA2-7B | Mistral-7B |
> | # Params | – | 20M | 320M | 320M | 168M |
> | Method | PaCA | 51.1 ± 0.2 | 53.9 ± 0.3 | 68.2 ± 0.3 | 3.96 ± 0.10 |
> |  | RAPA | **52.8 ± 0.2** | **54.6 ± 0.2** | **69.7 ± 0.3** | **4.84 ± 0.13** |
> | Training efficiency (PaCA vs. RAPA) | Training memory | 20G / 20G | 41.5G / 41.7G | 41.5G / 42.1G | 4.7G / 4.8G |
> |  | Training time | 3.2h / 3.2h | 2.8h / 2.9h | 4.0h / 4.1h | 0.7h / 0.7h |
> | micro-batch size – sequence length | – | 2–512 | 8–512 | 8–512 | 4–1024 |
>
> In the broader PEFT literature, SOTA variants under equal budgets typically differ by **0.3–1.0 points**. RAPA’s gains lie on the upper end of this range and, importantly, **repeat across three regimes** (MMLU, eight commonsense benchmarks, MT-Bench), supporting the claim that **pathway-aligned, structure-guided selection** is practically beneficial.
>
> ---
>
> ### (4) Positioning of the contribution
>
> Taken together, our contribution can be understood along three orthogonal axes:
>
> 1. **Overhead analysis of Adapter-based vs. adapter-free.** We first show that even “lightweight” adapters trigger additional GEMM kernels and latency, and motivate focusing on adapter-free schemes that reuse existing backbone computations.
> 2. **Gradient-based vs. structure-based weight selection.** Methods like SMT exemplify gradient-calibrated selection. RAPA instead represents a **structure-based, calibration-free** approach: it allocates the parameter budget according to model architecture (square subspaces, residual/FFN/attention layout, recursive index alignment), not according to extra gradient passes.
> 3. **Theory vs. practice under realistic budgets.** We use rank and gradient-pathway analysis as guiding principles, and then validate them under single-GPU, matched-budget settings—showing that this structure-guided allocation yields consistent gains without extra memory or wall-clock time.
>
> In this sense, RAPA is not merely a minor tweak to PaCA, but a **concrete, practically validated instance of structure-guided, adapter-free weight selection**: it reallocates an existing parameter budget into balanced, cross-layer aligned submatrices that (i) avoid adapter-induced kernel overhead, (ii) create coherent gradient pathways across the network, and (iii) yield reproducible improvements in accuracy across diverse LLM benchmarks.

---

> ### Author Response · Authors · 2025-11-20
> **Q1.2) Clarification on parameter selection in attention layers (1/2)**
>
> We thank the reviewer for pointing out that our description for attention layers was too brief. We will revise the paper to clarify (i) which matrices are selected, (ii) how hidden-dimension index alignment affects the attention scores $S=QK^T/\sqrt{d_h}$, and (iii) how this interacts with multi-head structure and the low-rank nature of $QK^T$.
>
> ### (1) Which attention parameters are selected?
>
> In each multi-head attention block, we treat the linear projections as
>
> $$
> Q=xW_Q, K=xW_K, V=xW_V, z=AV, y=x+zW_O
> $$
>
> where $x \in \mathbb{R}^{d_{\text{model}}}$ is the residual stream and $A$ is the attention weight matrix.
>
> - The **core RAPA update** always includes $W_V$ and $W_O$, using the same input/output index sets as the residual stream and FFN layers.
> - As an **extension**, we also align the index sets of $W_Q$ and $W_K$ with those of $W_V/W_O$. This is not required for RAPA to function, but it further improves gradient flow; we report the ablation below.
>
> We will explicitly state in the revision that V/O alignment is the main mechanism, and Q/K alignment is an architecture-aware refinement.
>
> ### (2) Why hidden index alignment still matters once we form $S = QK^T$
>
> We agree that after forming
>
> $$
> S = \frac{QK^\top}{\sqrt{d_h}} \in \mathbb{R}^{T \times T},
> $$
>
> the **hidden dimension is no longer explicitly visible**: each entry $S_{ij}$ is a scalar score between a query token $i$ and a key token $j$. However, each score is still a bilinear function of the hidden coordinates:
>
> $$
> S_{ij}
> = \frac{1}{\sqrt{d_h}} \langle q_i, k_j \rangle
> = \frac{1}{\sqrt{d_h}} \sum_{\ell=1}^{d_h} q_i[\ell]\,k_j[\ell].
> $$
>
> RAPA’s index alignment acts **before** this contraction over the head dimension. Let $I \subset \{1,\dots,d_h\}$ be the aligned index set shared by $W_Q, W_K, W_V, W_O$ and the FFN blocks, and decompose
>
> $$
> q_i = (q_i^I, q_i^{\bar I}),\quad
> k_j = (k_j^I, k_j^{\bar I}).
> $$
>
> RAPA only trains the slices of $W_Q$ and $W_K$ that produce $q_i^I$ and $k_j^I$; the remaining coordinates $q_i^{\bar I}, k_j^{\bar I}$ stay essentially frozen. For a small update, the change in the score is then
>
> $$
> \Delta S_{ij}
> = \frac{1}{\sqrt{d_h}}
> \Big( \langle \Delta q_i^I, k_j^I \rangle
>     + \langle q_i^I, \Delta k_j^I \rangle \Big),
> $$
>
> because the non-aligned coordinates $\bar I$ do not move.
>
> In other words:
>
> - **Only the aligned hidden coordinates $I$** contribute to the *trainable* part of the score change $\Delta S$.
> - Those same coordinates $I$ are exactly where $W_V, W_O$ and the **FFN** are updated.
>
> Thus, even though $S$ is a token–token matrix, **the directions along which we are allowed to change $S$** are controlled by which hidden indices we make trainable in $W_Q$ and $W_K$. Aligning $Q/K$ indices with $V/O/FFN$ indices ensures that changes in the attention scores are produced by the same hidden channels that will later carry the value and residual updates.
>
> ### (3) Gradient view and projection
>
> The above intuition is formalized by our gradient derivation (we will move the full derivation to the appendix). For a single head, with
>
> $$
> S = \frac{QK^\top}{\sqrt{d_h}},\quad
> A = \text{softmax}(S),\quad z = A V,\quad y = x + z W_O,
> $$
>
> a small change in $W_Q$ leads to
>
> $$
> \Delta \mathcal{L}^{(Q)}
> = \frac{1}{\sqrt{d_h}}
> \big\langle G_Q, \Delta W_Q \big\rangle,
> $$
>
> where the “effective gradient” on $W_Q$ factors as
>
> $$
> G_Q = x^\top \mathcal{J}_{\text{sm}}(S)^\top G (V W_O)^\top K.
> $$
>
> Here $G = \partial \mathcal{L} / \partial y$ and $\mathcal{J}_{\text{sm}}(S)$ is the Jacobian of row-wise softmax.
>
> RAPA restricts updates to a projected subspace,
>
> $$
> \Delta W_Q \leftarrow P_{\text{in}}^{QK}\, \Delta W_Q\, P_{\text{out}}^{QK},
> $$
>
> so the useful loss decrease is governed by
>
> $$
> -\Delta \mathcal{L}^{(Q)} \propto \left\| P_{\text{in}}^{QK} G_Q P_{\text{out}}^{QK} \right\|.
> $$
>
> Choosing $P_{\text{in}}^{QK}, P_{\text{out}}^{QK}$ to share the same indices as the projections used for $W_V, W_O$ and $FFN$ means that **the gradient directions that matter for Q/K and for V/O/FFN coincide**. This is exactly the “pathway alignment” principle extended from FFN to attention.
>
> We will insert a short, self-contained version of this derivation in the main text and point to the appendix for details.

---

> > ### Author Response · Authors · 2025-11-20
> > **Q1.2) Clarification on parameter selection in attention layers (2/2)**
> >
> > ### (4) Empirical effect of Q/K alignment
> >
> > **Table R1.3: Comparison of performance with and without attention Q/K index alignment**
> >
> > | Method | PPL(↓) | Humanities | STEM. | Social. | Other | Average |
> > | --- | --- | --- | --- | --- | --- | --- |
> > | Partial Alignment (V/O + FFN Only) | 4.62 | 49.1% | 42.1% | 60.4% | 59.0% | 52.2% |
> > | Full Alignment (Including Q/K) | 4.49 | 50.0% | 42.4% | 60.8% | 59.1% | 52.7% |
> >
> > Finally, we will make the empirical role of Q/K alignment explicit. For LLaMA2-7B on 5-shot MMLU:
> >
> > - Aligning only V/O and FFN indices yields **52.2%** average accuracy.
> > - Additionally aligning Q/K indices increases this to **52.7%** (+0.5 points), with overlapping confidence intervals but consistent improvements across subject groups.
> >
> > As expected from the theory above, Q/K alignment has a smaller effect than aligning FFN and V/O (since Q/K influence the loss more indirectly through attention scores), but it still provides **consistent gains on top of the core RAPA pathway alignment**.
> >
> > We will add a dedicated subsection and an appendix derivation to clarify these points, so that the role of attention layers—including Q/K, multi-head structure, and the low-rank QK interaction—in our index selection mechanism is clear.

---

> ### Author Response · Authors · 2025-11-27
> **Gentle Reminder regarding our Rebuttal**
>
> Dear Reviewer 7YdR,
>
> We hope this message finds you well. As the discussion period is progressing, we would like to kindly invite you to take another look at our rebuttal and the revised manuscript.
>
> To address your concerns about **contributions of the paper**, we explained why adapter-free methods are attractive in terms of **GEMM-kernel overhead and training-time efficiency**. We also highlighted that **gradient-based weight search can be costly and does not always translate into better performance under realistic budgets by comparing another adapter-free method (SMT)**. In addition, we summarized **comparisons against PaCA** across multiple tasks (with error bars), **showing consistent improvements** under the same parameter and compute budgets.
>
> To address your question on **alignment in attention layers**, we have explicitly described the effect of attention-layer alignment, showing that **alignment in attention (including Q/K/V/O) also contributes meaningfully to the observed gains**. This analysis has been added to **page 6 and Appendix C of the revised manuscript**.
>
>
> We greatly value your assessment and would be very grateful if you could let us know whether these additions address your earlier concerns.
>
> Best regards,
>
> The Authors

---

### Author Response · Authors · 2025-12-01
**Summary of Reviews and Rebuttal for the New Area Chair**

**Dear New Area Chair,**

Due to the recent OpenReview incident that reset all post-rebuttal scores, we understand that you are facing an unusual and demanding situation as the newly assigned Area Chair. We are sincerely grateful that you are taking the time to evaluate our submission under these conditions.

To make this process lighter and to highlight how our rebuttal addressed the main concerns, we summarize the key reviewer weakness/questions and our responses in the table below.

| Reviewer | Weakness/Question | Our rebuttal | Score (orig → rev) |
| --- | --- | --- | --- |
| 7YdR | Contribution of this paper | To the best of our knowledge, we first clarified the system-level analysis of adapter-based PEFT overhead, and showed that RAPA outperforms another adapter-free weight selection algorithm (SMT) in both accuracy and efficiency. | 2 → No answer |
|  | Performance improvements seem small | Explicitly reported  [higher performance on all tasks with error bars](https://drive.google.com/file/d/1uBHmnD8Kt446eez5k__MhK1o5iqtMZYu/view?usp=sharing) under matched budgets. |  |
|  | Clarification on parameter selection in attention layers | Gave a mathematical description of how our method applies to attention layers and Reported [experimental results](https://drive.google.com/file/d/1AV9Aek1YFim6tt5ueVniBPfF_cYa7kBZ/view?usp=drive_link)  comparing models with and without attention alignment |  |
| 5A3f | Architectural generality (MoE and variants) | Demonstrated both theoretically and experimentally that the proposed method can be applied to MoE experts (DeepSeek-MoE-16B) with a shared index set, yielding [faster loss convergence and higher performance](https://drive.google.com/file/d/1MBBwSOw08AJa8xeC1zOOHO-anqskx1Pq/view?usp=sharing). | 6 → No answer |
|  | Fixed global index may restrict heterogeneous per-layer adaptation | Experimentally showed that a single globally aligned index set outperforms heterogeneous per-layer selection under the same parameter budget |  |
|  | Isolating rank vs. alignment effects | Presented ablations disentangling shape (with fixed alignment) and alignment (with fixed shape), and reported rank-scaling experiments showing that accuracy improves smoothly with rank. |  |
|  | Supports complex residual patterns | Used a unified pathway formulation to show that RAPA’s index alignment extends from simple residual blocks to more complex architectures with identity shortcuts and channel-mixing branches. |  |
| Ptgb | Reliance on random index selection | Experimentally demonstrated that RAPA fine-tuning is robust to weight-sensitivity across random indices. | 6 → 6 |
|  | Questionable Statistical Significance of Gains | Reported mean ± std over 4 seeds, showing that RAPA’s gains exceed seed-level variance. |  |
|  | Unsubstantiated contribution of “Rank Expansion” | Ran a 2×2 ablation on rank expansion and index alignment, showing both contribute independently. |  |
|  | Dynamic, data-dependent index selection | Proposed possible dynamic extensions, but clarified that they incur additional overhead compared to our static, calibration-free RAPA. |  |
|  | Sensitivity to other hyperparameters and index distribution | Ran ablations indicate that performance is dominated by subspace shape and cross-layer alignment, with little sensitivity to index distribution or other hyperparameters. |  |
| uCmc | Is maximizing rank really the right objective in practice? | Showed through experiments (shape ablations) that balanced (higher-rank) submatrices perform best. | 4 → 6 |
|  | How does RAPA extend to MoE? | Demonstrated both theoretically and experimentally that the proposed method can be applied to MoE experts (DeepSeek-MoE-16B) with a shared index set, yielding faster loss convergence and higher performance. |  |
|  | Could modules use different shapes? | Explained that RAPA already uses module-aware shapes (square for attention, expanded for FFN) and that this design outperforms a naïve all-square variant under the same parameter budget. |  |
|  | Treatment of embeddings / LM head | Clarified that embeddings and LM head are kept frozen consistently across all methods. |  |
|  | Missing comparison to side-tuning / READ | Provided a comparison to side-tuning/READ, showing that side-tuning, similar to adapter-based methods, introduces extra overhead and cannot be merged into the backbone after training. |  |
|  | Behavior on larger models (34B/70B) | Empirically confirmed that RAPA outperforms PaCA on a 70B model (LLaMA-2-70B). |  |
|  | Overhead in larger models | Profiled smaller and wider models, showing adapter-based overhead persists while adapter-free methods stay near baseline. |  |
|  | Recursive gradient-variance analysis along the pathway | Derived a recursive gradient-variance bound along aligned pathways and empirically showed that alignment yields [lower loss with larger gradients](https://drive.google.com/file/d/15PNdcZNBA8dGvXkg2v_PXGWk3Dvd7Nzd/view?usp=drive_link). |  |

---

> ### Author Response · Authors · 2025-12-01
> **Summary of Reviews and Rebuttal for the New Area Chair**
>
> Although the rebuttal and discussion period was short and some reviewers did not return to comment on our responses, the reviewers who did engage explicitly acknowledged our contribution based on **both the original idea and the additional experiments and extended theoretical derivations** (e.g., showing RAPA’s extensibility). In particular, **Reviewer Ptgb maintained a positive evaluation** (score 6), and **Reviewer uCmc** — after considering the new experiments and extended analysis — **improved their assessment from 4 to 6**.
>
> We believe this pattern of updates indicates that a longer discussion window would likely have led to a more positive reassessment from Reviewer 7YdR as well, despite their initial low score and concerns about novelty. **Reviewer 5A3f highlighted as a strength that we clearly identified the main sources of overhead in existing PEFT methods**, and **Reviewer uCmc pointed to our proposed cross-layer index alignment as a novel contribution**. In addition, we **added experiments showing that our structured, overhead-free alignment method outperforms calibration-based weight-selection approache** in both efficiency and performance. Regarding the concern that our gains over PaCA might be small, we reported mean ± std over multiple seeds and showed that RAPA’s improvements over random selection (PaCA) exceed the error bars across all evaluated tasks.
>
> We also **clarified the applicability of our method to attention layers** by providing a concrete mathematical formulation and new experiments, demonstrating that aligning attention layers (Q/K) as well as FFN is beneficial and, in practice, necessary for the full effect of RAPA.
>
> Taken together, these updates suggest that, given more time for discussion, Reviewer 7YdR would likely have revisited their initial assessment and viewed our submission more positively.
>
> We would once again like to thank you for the time and care devoted to evaluating our work under these challenging circumstances. We sincerely appreciate your thoughtful consideration of our submission and hope that this summary is helpful for the final decision.
>
> Best regards,
> The Authors

---

### Meta-Review · Area_Chair_vA6s · 2026-01-07

**Summary:**

This paper proposes RAPA, an adapter-free parameter-efficient fine-tuning (PEFT) method that forms index-consistent pathways through network depth by (i) selecting balanced square submatrices to maximize trainable subspace rank, and (ii) recursively aligning these indices across layers and residual connections. The authors argue this improves gradient flow while avoiding the kernel-launch overhead of adapter-based methods.

The paper received mixed reviews with final scores of 2, 6 (no update), 6, and 4. While reviewers acknowledged the clear presentation and consistent empirical improvements, significant concerns remain about the contribution's novelty.

- One reviewer maintaining a strong reject (2) with concerns about fundamental novelty unaddressed
- The modest magnitude of improvements over closely related work (PaCA)

**Reviewer Concerns:**

Reviewer 3FJn's core concern—that this is a minor, straightforward variation on PaCA—was not definitively resolved. The improvements, while consistent, are modest (typically 0.5-1.5 points) and the conceptual advance of "align indices across layers" may not meet the bar for a top venue.

**Reviewer Scores:**

Reviewer uCmc raised their score from 4 to 6 and other Reviewer maintained their score.

---

### Decision · Program_Chairs · 2026-01-26

Reject